# A deep unrolled neural network for real-time MRI-guided brain intervention

Zhao He [1,2,3], Ya-Nan Zhu[4], Yu Chen[1,2,3], Yi Chen[1,2,3], Yuchen He[5], Yuhao Sun[6], Tao Wang[6], Chengcheng Zhang [6], Bomin Sun [6], Fuhua Yan[7], Xiaoqun Zhang[4,8], Qing-Fang Sun [6] ✉, Guang-Zhong Yang [1,2,3] ✉ & Yuan Feng [1,2,3,7] ✉

Accurate navigation and targeting are critical for neurological interventions including biopsy and deep brain stimulation. Real-time image guidance further improves surgical planning and MRI is ideally suited for both pre- and intra-operative imaging. However, balancing spatial and temporal resolution is a major challenge for real-time interventional MRI (i-MRI). Here, we proposed a deep unrolled neural network, dubbed as LSFP-Net, for real-time i-MRI reconstruction. By integrating LSFP-Net and a custom-designed, MR-compatible interventional device into a 3 T MRI scanner, a real-time MRI-guided brain intervention system is proposed. The performance of the system was evaluated using phantom and cadaver studies. 2D/3D real-time i-MRI was achieved with temporal resolutions of 80/732.8 ms, latencies of 0.4/3.66 s including data communication, processing and reconstruction time, and in-plane spatial resolution of $1 \times 1$ mm$^2$. The results demonstrated that the proposed method enables real-time monitoring of the remote-controlled brain intervention, and showed the potential to be readily integrated into diagnostic scanners for image-guided neurosurgery.

Real-time guidance and visualization are crucial for robot-assisted surgery, especially in neurosurgery, where accurate positioning and delicate interventions are essential[1,2]. These include brain biopsy[3], ablation[4], and electrode placement in Deep Brain Stimulation (DBS)[5]. Computed tomography (CT) and ultrasound are commonly used for navigation for robot-assisted neurosurgery, due to their fast imaging capabilities. However, CT has ionizing radiation, and ultrasound cannot provide enough resolution and tissue contrast for the brain. Magnetic Resonance Imaging (MRI), devoid of ionizing radiation but with superior soft-tissue contrast is ideal for brain interventional

imaging[6,7]. However, the relatively slow imaging speed poses a significant hurdle for clinical applications.

Balancing temporal and spatial resolution is a major challenge for real-time interventional MRI (i-MRI) in neurosurgery[8,9]. Acceleration techniques such as balanced steady-state precession (bSSFP)[10], parallel imaging[11], generalized series[12], and keyhole imaging[13] are difficult to meet real-time requirements. Over the past decade, compressed sensing (CS)-based methods exploiting data sparsity have been used to accelerate imaging speed[14,15]. By further utilizing the temporal information, CS-based k-t methods have been proposed for real-time

[1]School of Biomedical Engineering, Shanghai Jiao Tong University, Shanghai 200030, China. [2]Institute of Medical Robotics, Shanghai Jiao Tong University, Shanghai 200240, China. [3]National Engineering Research Center of Advanced Magnetic Resonance Technologies for Diagnosis and Therapy (NERC-AMRT), School of Biomedical Engineering, Shanghai Jiao Tong University, Shanghai 200240, China. [4]School of Mathematical Sciences, MOE-LSC and Institute of Natural Sciences, Shanghai Jiao Tong University, Shanghai 200240, China. [5]Department of Mathematics, City University of Hong Kong, Kowloon, Hong Kong SAR. [6]Department of Neurosurgery, Ruijin Hospital affiliated to Shanghai Jiao Tong University School of Medicine, Shanghai 200025, China. [7]Department of Radiology, Ruijin Hospital affiliated to Shanghai Jiao Tong University School of Medicine, Shanghai 200025, China. [8]Shanghai Artificial Intelligence Laboratory, Shanghai 200232, China. ✉e-mail: rjns123@163.com; gzyang@sjtu.edu.cn; fengyuan@sjtu.edu.cn

dynamic MRI[16–18]. Low-rank matrix imaging involving one or more dimensions has become an established method for fast MR imaging[19–22]. Decomposing the data matrix into a low-rank component (L) and a sparse component (S), i.e., low-rank plus sparse decomposition (L + S) or robust principle component analysis (RPCA), has been proposed and applied for dynamic MRI[23–26] and other fields[27,28]. In addition, high temporal resolution imaging has been achieved using a nonlinear inverse reconstruction method with undersampled radial sampling for real-time cardiovascular MRI[29,30]. However, the relatively low spatial resolution may not satisfy the requirements for neurosurgery. Recently, a combination of Low-rank and Sparsity decomposition with Framelet transform and Primal dual fixed point optimization (LSFP) was proposed for i-MRI reconstruction with a group-based reconstruction scheme[31]. However, complex parameter tuning and long computation time are required, which cannot satisfy the online reconstruction requirement of real-time i-MRI.

Deep learning (DL) can greatly improve reconstruction quality and accelerate computation speed, making it especially useful in fast MRI[32–34]. Typical DL networks include AUTOMAP[35], GAN[36], U-nets[37], transformers[38], and diffusion models[39]. For further utilizing temporal information, a convolutional recurrent neural network (CRNN) was proposed for dynamic MRI[40]. Similarly, Jaubert et al. developed a deep artifact suppression method using recurrent U-Nets for real-time cardiac MRI[37]. A DL-based image reconstruction and motion estimation from undersampled radial k-space has also been applied to real-time MRI-guided radiotherapy[41]. However, these data-driven networks rely on large-scale training datasets and have limited interpretability and generalizability[42]. To overcome these limitations, unrolled networks were proposed. Typical models include cascaded networks[43], ISTA-Net (unrolling of the iterative shrinkage-thresholding algorithm)[44], ADMM-Net (unrolling of alternating direction method of multipliers method)[45], and variational network (unrolling of gradient descent algorithm)[46]. An unrolled variational network with an undersampled spiral k-space trajectory was also developed for real-time cardiac MRI reconstruction[47]. However, only exploiting sparse prior limits the performance of these networks. By utilizing both low-rank and sparse priors, SLR-Net[48] and L + S-Net[49] have become two state-of-the-art

unrolled networks for dynamic MRI reconstruction. However, SLR-Net and L + S-Net are designed for retrospective reconstruction with Cartesian sampling, which cannot satisfy the online reconstruction requirement of real-time i-MRI.

In this study, we proposed LSFP-Net for i-MRI reconstruction by unrolling the iterative LSFP algorithm into a neural network. The low-rank and sparse priors and spatial sparsity of both low-rank and sparse components are utilized. The group-based reconstruction with periodic radial sampling makes LSFP-Net satisfy the online reconstruction requirement for real-time i-MRI. Simulated and clinical images were used to train and test the LSFP-Net. By deploying the trained LSFP-Net on a 3 T MRI scanner, we used a custom-designed interventional device to demonstrate the feasibility of the proposed i-MRI system for brain intervention. Imaging performance was evaluated using interventional phantoms and cadaver studies.

## Results

### Real-time i-MRI reconstruction with LSFP-Net

In i-MRI, the slowly changing background and dynamic interventional features could be decomposed into a low-rank matrix **L** and a sparse matrix **S** from an image sequence $\mathbf{x}$[31], i.e., $\mathbf{x} = \mathbf{L} + \mathbf{S}$. A Low-rank and Sparsity decomposition with Framelet and Primal dual fixed point (LSFP) method for i-MRI reconstruction was unrolled into a deep neural network, dubbed as LSFP-Net, for real-time i-MRI reconstruction (Supplementary Fig. 1 and Supplementary Movie 1). Specifically, LSFP-Net is composed of $N_b$ blocks and each block strictly corresponds to one iteration in the LSFP algorithm. In LSFP-Net, fewer iterations were needed, which can significantly reduce the reconstruction cost. The hyper-parameters are learnable during the LSFP-Net training, which avoids tedious parameter tuning.

To improve motion resilience and achieve high-fold acceleration, the k-space data were acquired with a multi-coil golden-angle radial sampling, and the sampling trajectory was repeated with a period of one group of spokes (Fig. 1a). Multiple interventional images were simultaneously reconstructed in a group-wise manner with low-rank and sparse constraints along the temporal dimension (Fig. 1b). A simulated interventional dataset was prepared to train LSFP-Net

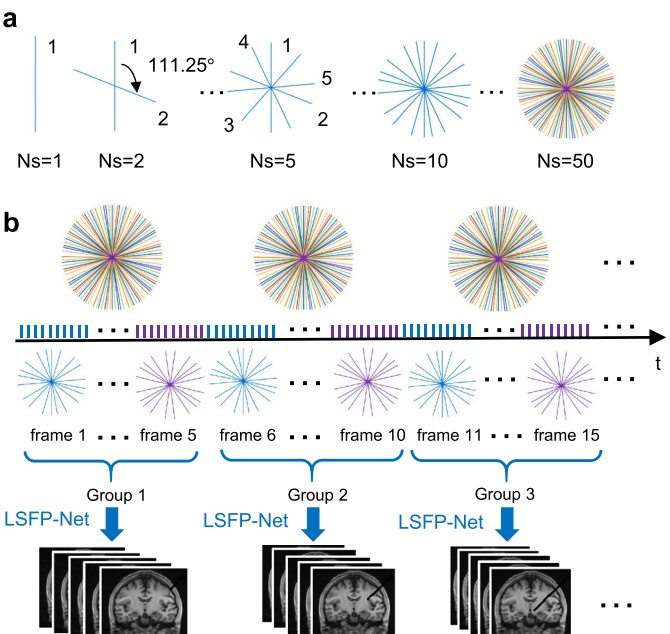

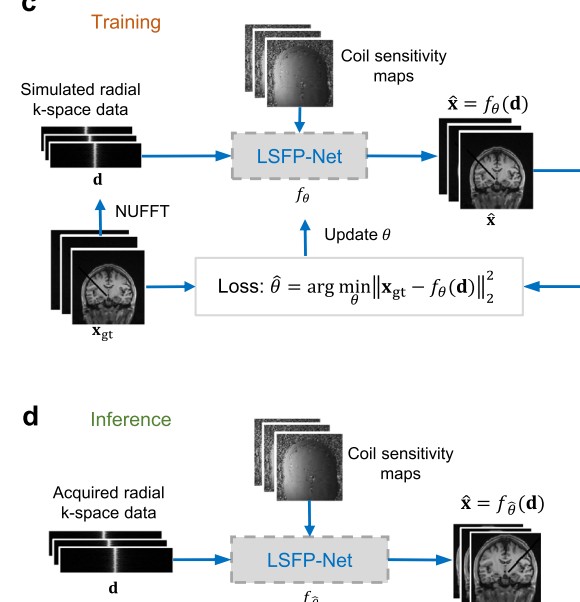

**Fig. 1 | LSFP-Net for real-time i-MRI reconstruction. a** A golden-angle radial sampling pattern (golden angle = 111.25°) was used for k-space data acquisition. **b** Multiple interventional images were reconstructed simultaneously in a group-wise way. **c** LSFP-Net was trained on a simulated dataset. **d** For inference, interventional images were reconstructed in real-time using the trained LSFP-Net.

(Fig. 1c). For inference, interventional images were reconstructed in real-time using the trained LSFP-Net (Fig. 1d).

For 3D imaging, a stack-of-stars golden-angle radial sampling scheme was adopted (Supplementary Fig. 2a). This allows for the different resolutions in the x-y plane and in the z direction, and a slice-by-slice 2D reconstruction uses less memory[50]. The sampling trajectory was also repeated with a period of one group of spokes. After collecting one group of data, all slices were detangled by Fast Fourier Transform (FFT) along the z direction. Then, the k-space data of each slice was divided into several frames and reconstructed by LSFP-Net (Supplementary Fig. 2b).

## Performance evaluation on the simulated dataset of brain intervention

For training of the LSFP-Net, brain intervention was simulated based on brain images acquired from volunteers (Fig. 2a and Supplementary Movie 2). With 10 spokes for the reconstruction of each frame (acceleration factor $R = 20$), we compared the results of LSFP-Net with two iterative CS methods (L + S[23] and LSFP[31]) and four DL-based methods (CRNN[40], ISTA-Net[44], SLR-Net[48], and L + S-Net[49]). Compared to other methods, LSFP-Net had the fewest artifacts and errors and perfectly recovered the interventional features (Fig. 2b, c, and Supplementary Movie 3). Quantitatively, LSFP-Net yielded the near-highest

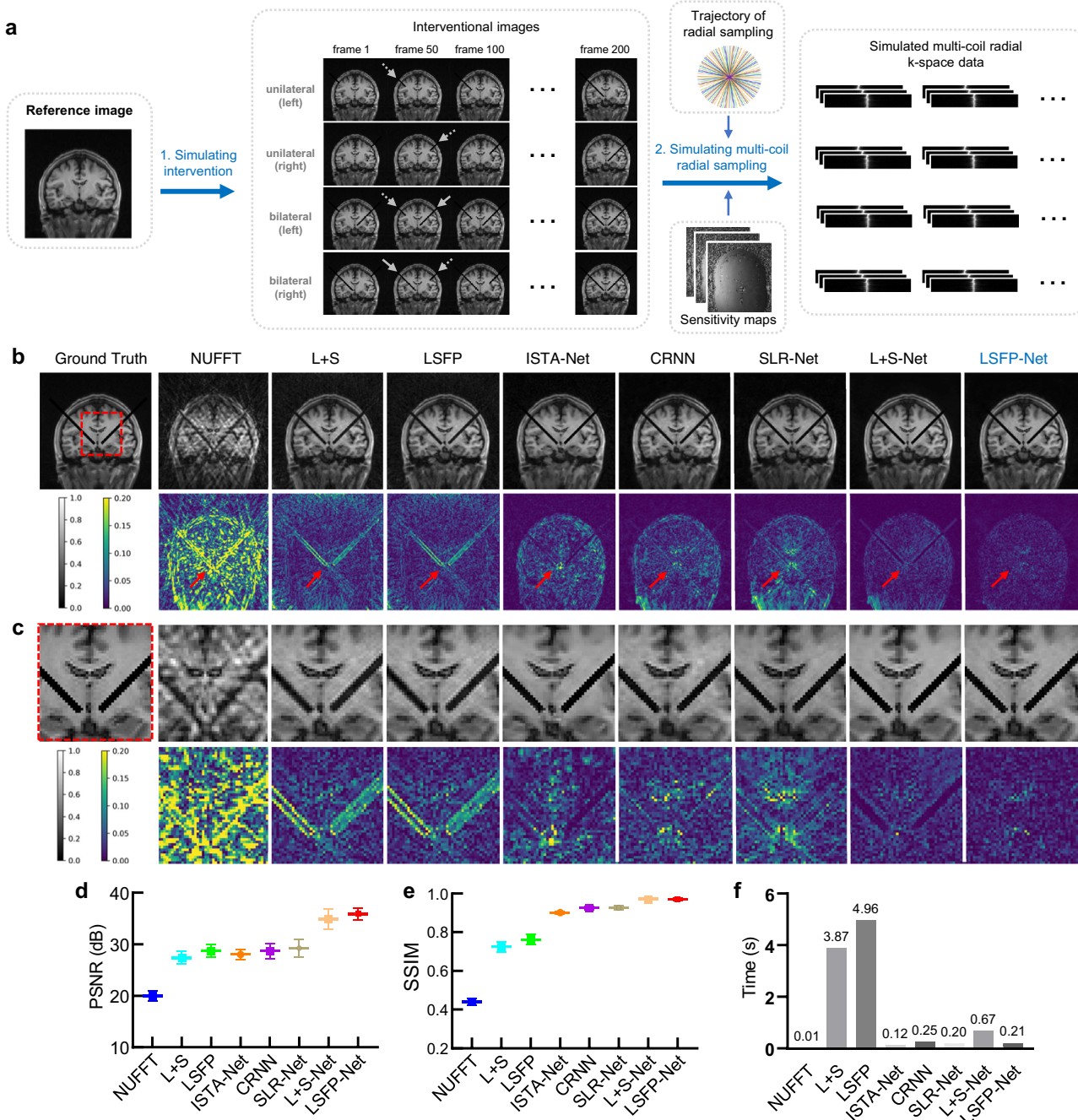

**Fig. 2 | The preparation of the training dataset and a comparison of different methods. a** A simulated dataset of brain intervention was prepared. **b** A comparison of different methods. 10 spokes were used for the reconstruction of each frame (acceleration factor $R = 20$). Five frames per group were used for L + S, LSFP, CRNN, SLR-Net, L + S-Net, and LSFP-Net. **c** The magnified view of the interventional features. The area of the interventional feature is indicated by the red dashed box in **b**. **d–f** The quantitative metrics of the different methods. The pixel values were normalized without dimension. **d, e** The data are presented as mean values ± standard deviation, and the sample size is $n = 64$. Source data of **d–f** are provided as a Source Data file.

PSNR and SSIM (Fig. 2d, e). In terms of reconstruction time, LSFP-Net required only 0.21 s for one group reconstruction, which is comparable to other DL-based methods (CRNN, ISTA-Net, SLR-Net, and L + S-Net) and about 10 times faster than CS-based methods (L + S and LSFP) (Fig. 2f). The enhanced performance of LSFP-Net can be attributed to its ability to leverage the spatial sparsity of both low-rank and sparse components, as well as its capability to learn the regularized parameters during the training stage, setting it apart from other methods.

### Acceleration factors

A total of 5, 8, and 20 spokes for each frame were used for reconstruction to evaluate the performance of the proposed method (Fig. 3a–c). The acceleration factors are 40, 25, and 10, respectively. The proposed LSFP-Net also achieved a near-optimal performance in terms of PSNR and SSIM. The reconstruction time for both the LSFP-Net and L + S-Net is less than one second. The computation speed of LSFP-Net is more than 20 times faster than that of LSFP, and approximately 3 times faster than that of L + S-Net.

### Iterations and convolutional layers for LSFP-Net

The numbers of iterative blocks $N_b$ and convolutional layers $N_c$ in $\{\psi_L, \psi_L^T, \psi_S, \psi_S^T\}$ determine the depth of LSFP-Net. To figure out their effect on the reconstruction, different values of $N_b$ and $N_c$ were investigated.

First, different $N_b = 1, 2, 3, 5, 7, 9$, and 11 were used with a fixed $N_c = 3$ (Fig. 3d–f). The PSNR/SSIM improved from 28.42/0.93 to 39.11/0.99 with the increase of iteration blocks from 1 to 11, and the increase of time cost from 0.11 s to 1.02 s. Therefore, a tradeoff is needed to balance performance and computational time. A comparison with other DL-based methods showed that LSFP-Net with $N_b = 2$ achieved better performance with a similar number of parameters (Fig. 2 and 3, and Supplementary Table 1). Moreover, when $N_b$ was increased to 3, the PSNR/SSIM values improved with only a slight increase of 0.09 s in computational time. As a result, $N_b = 3$ was chosen for the phantom and cadaver experiments. Second, $N_c = 3, 5, 7, 9$, and 11 were also used with a fixed $N_b = 3$ (Fig. 3g–i). The reconstruction quality fluctuated and the reconstruction time increased with increasing $N_c$. This was because LSFP-Net became more complex as the number of convolutional layers increased, which could affect the generalizability of the network. Therefore, $N_b = 3$ and $N_c = 3$ were selected for the reconstruction of phantom and cadaver experiments.

### Parameters of LSFP-Net

For real-time imaging, a group-based data acquisition and reconstruction scheme was adopted. For $p$ radial spokes of k-space data, we can use different combinations of $m$ spokes per frame (SPF = $m$) and $n$ frames per group (FPG = $n$) for reconstruction when satisfying

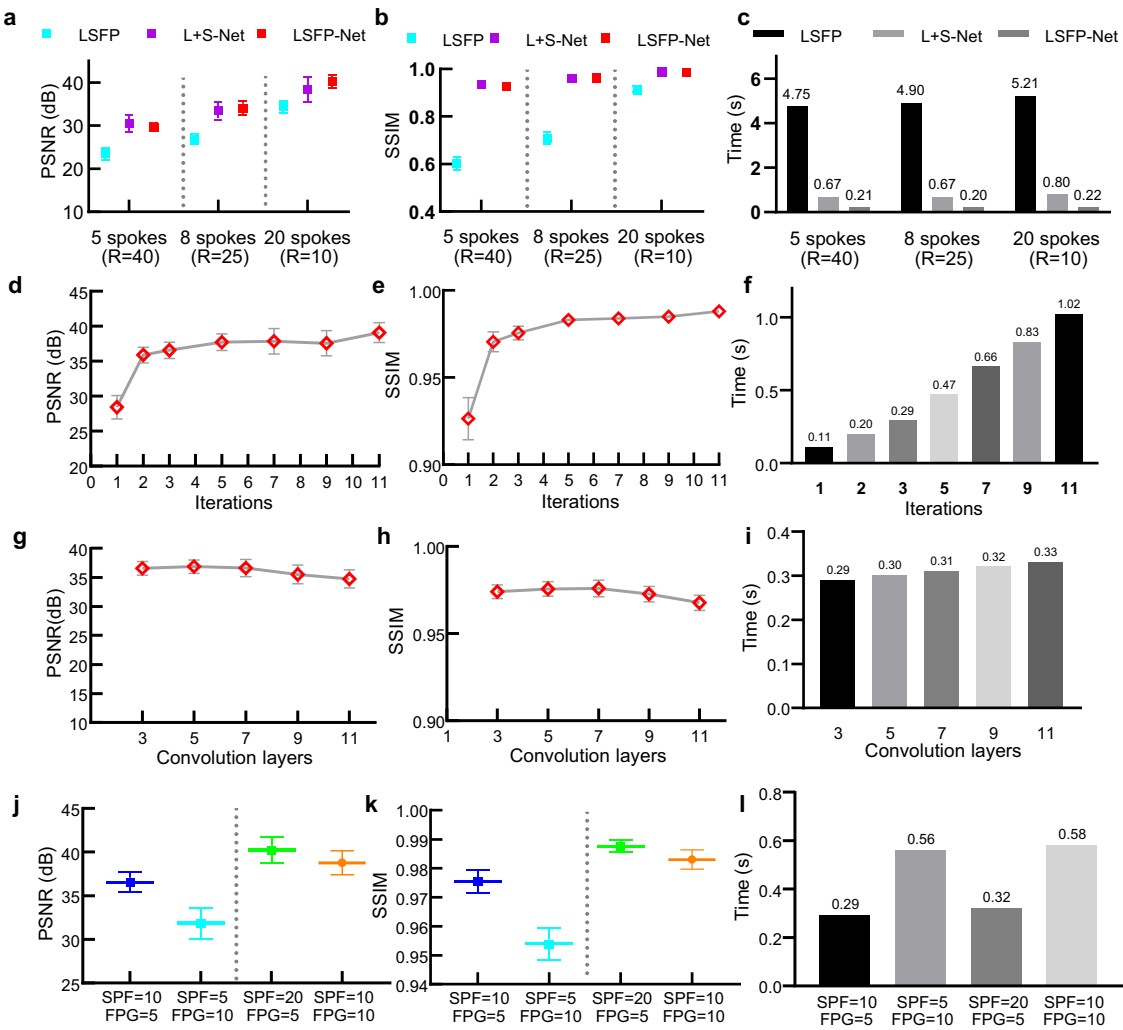

**Fig. 3 | Results of different parameters for LSFP-Net. a–c** A comparison of different methods with $R = 40, 25$, and 10. **d–f** The results of different iterations of LSFP-Net. **g–i** The results of different convolution layers for the sparsity transform of LSFP-Net. **j–l** The results of different combinations of spokes per frame (SPF) and frames per group (FPG). **a, b, g, h, j, k** The data are presented as mean values ± standard deviation, and the sample size is $n = 64$. Source data of **a–l** are provided as a Source Data file.

$p = m \times n$. Here, we reconstructed images using two combinations of (SPF = 10, FPG = 5) and (SPF = 5, FPG = 10) for 50 radial spokes and using two combinations of (SPF = 20, FPG = 5) and (SPF = 10, FPG = 10) for 100 radial spokes (Fig. 3j–l). A higher SPF could result in better reconstruction quality but lower temporal resolution per frame. A smaller SPF may result in a higher temporal resolution per frame but a longer reconstruction time due to a higher FPG. Therefore, a tradeoff between temporal resolution and reconstruction quality can be achieved by selecting different combinations of SPF and FPG. Here considering the reconstruction quality, we chose (SPF = 20, FPG = 5) for the reconstruction of phantom and cadaver experiments.

### Performance evaluation on the simulated dataset of DBS electrode placement

Based on a series of postoperative MR images of DBS electrode placement, a simulated dataset was generated to evaluate the generalizability of LSFP-Net (Fig. 4a). Compared to other methods, there are no obvious artifacts in the images reconstructed by L + S-Net and LSFP-Net, and the interventional features can be easily distinguished (Fig. 4b). In terms of PSNR and SSIM, the performance of LSFP-Net (PSNR/SSIM = 28.45/0.92) was comparable to that of L + S-Net (PSNR/SSIM = 28.25/0.95). The reconstruction speed of LSFP-Net (0.23 s) was about 5 times faster than that of L + S-Net (1.30 s), outperforming the iterative methods (L + S and LSFP).

### Real-time MRI-guided brain intervention system

To achieve the brain intervention with real-time i-MRI guidance, an i-MRI system integrating LSFP-Net and a custom-designed interventional device was implemented on a 3 T MRI scanner (uMR 790, United Imaging Healthcare, Shanghai, China) (Fig. 5a). A trained LSFP-Net was deployed on a Gadgetron server. Continuously acquired radial k-space data during the intervention was reconstructed online at the server and immediately sent to the console. The custom-built interventional

device was fixed on the skull for intervention. The interventional device was connected to a stepper motor via torque rods and a flexible shaft. The interventional depth and speed were controlled by the motor in the control room.

An MR-compatible interventional device with 4 degrees of freedom (DOFs) was designed. The device consisted of a base, a ball-joint, a locking ring, and a lead screw-nut mechanism (Fig. 5b). The ceramic interventional needle (φ1.5 mm) was fixed on the nut. Figure 5c shows the system components in the control room, including the stepper motor, driver, controller, and power supply. Figure 5d, e show the components in the MR scanner room, including the interventional device and the torque rod. The MR compatibility testing of the interventional device showed that the presence of the device had no effect on MR imaging (Supplementary Fig. 5).

### Phantom experiments

Two phantom intervention experiments were carried out to test the real-time imaging and remotely actuated intervention capabilities of the brain intervention system. In the fruit phantom, several red cherry tomatoes and green grapes were embedded in a gelatin-filled cylinder (Fig. 6a). The interventional device was fixed on the cover of the cylinder and placed in the MR head coil (Fig. 6b). In the porcine-brain phantom, two porcine brains were embedded into a gelatin-filled 3D-printed human-skull model (Fig. 6c). An interventional device based on a lead-screw mechanism was placed in front of the phantom (Fig. 6d).

Before and after the intervention, two fully sampled 2D images (1.6 s/frame) were acquired to show the initial and final positions of the needle (Fig. 6e, g). For the fruit phantom, the ceramic needle moved ~61 mm with a uniform velocity of ~1.525 mm/s for 40 s. For the porcine-brain phantom, the ceramic needle moved ~90 mm with a uniform velocity of ~2.25 mm/s for 40 s. 2D radial k-space data were acquired with a temporal resolution of 80 ms/frame (400 ms/group). LSFP-Net reconstructed images in real-time for 370 ms/group (Fig. 6f,

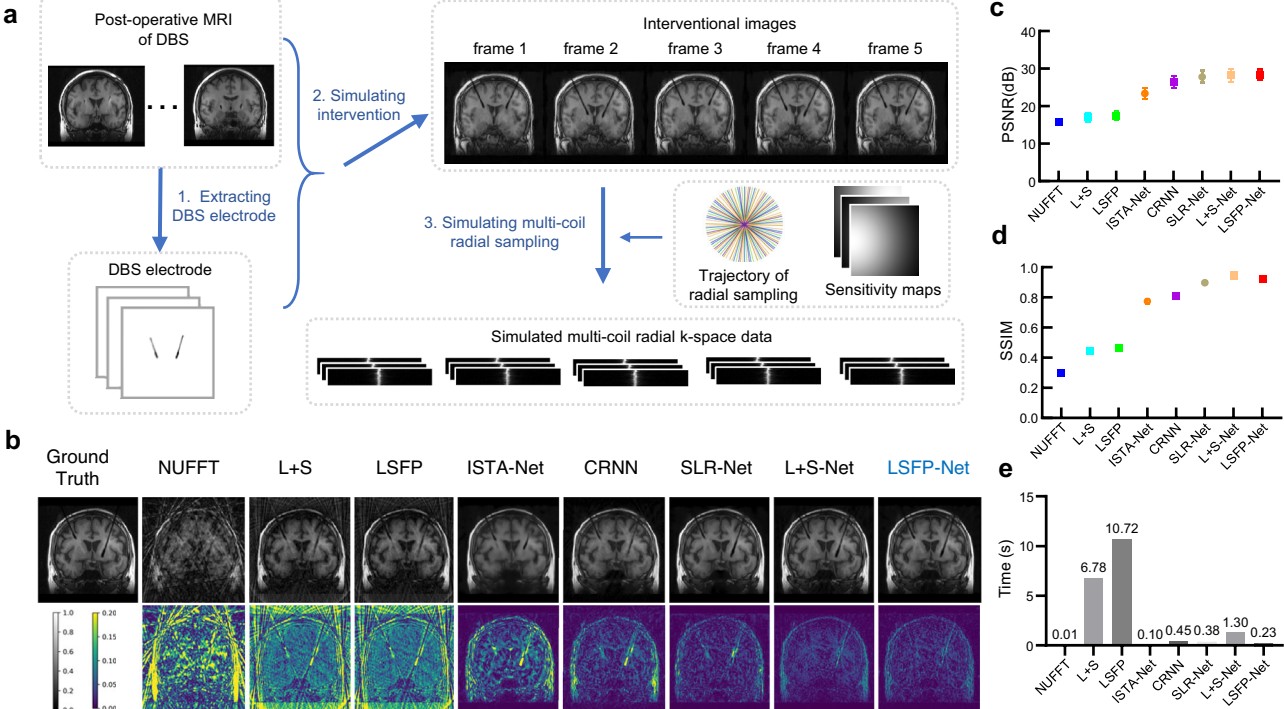

**Fig. 4 | A comparison of different methods on the simulated DBS electrode placement dataset. a** The dataset preparation is based on the postoperative MRI of DBS electrode placement. **b** The reconstruction results from different methods using 10 spokes. **c**–**d** The quantitative metrics of different methods. The pixel

values were normalized without dimension. **c** the data are presented as mean values ± standard deviation, and the sample size is $n = 188$. Source data of **c**–**e** are provided as a Source Data file.

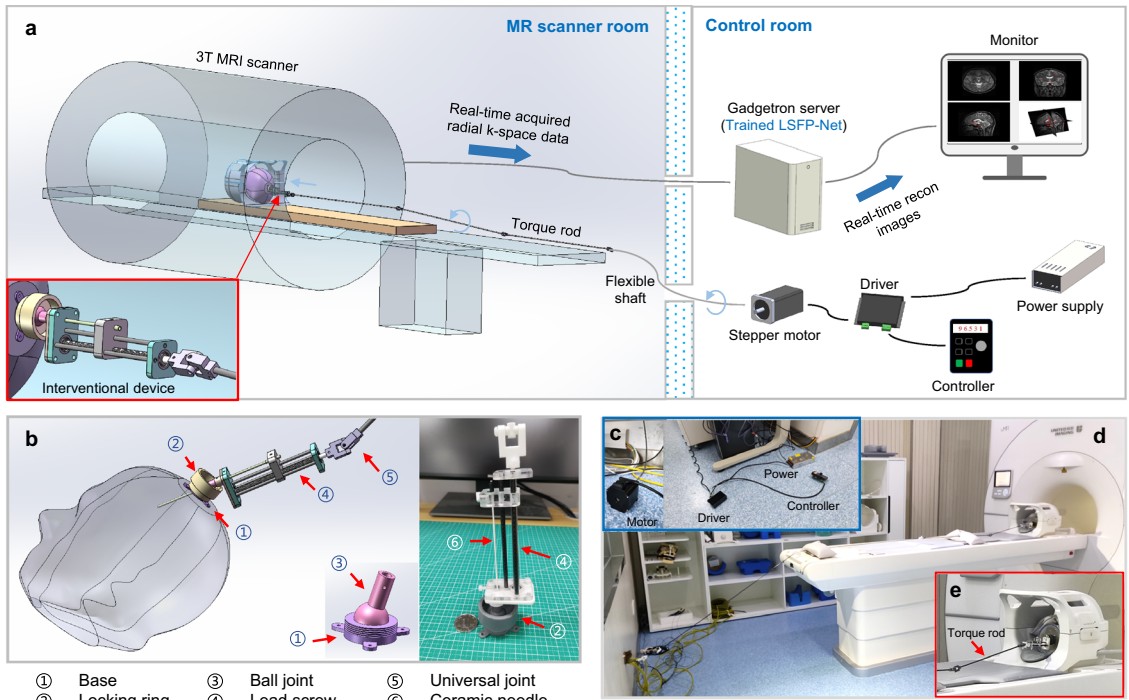

① Base          ③ Ball joint        ⑤ Universal joint
② Locking ring  ④ Lead screw        ⑥ Ceramic needle

**Fig. 5 | Real-time MRI-guided brain intervention system. a** The MR-compatible components of the system were located in the MR scanner room. The ferromagnetic components were located in the control room. **b** The custom-designed interventional device. **c** The components in the control room. **d** The components in the MR scanner room. **e** The partially magnified view of **d**.

Supplementary Movie 4, and Supplementary Movie 5). In the fruit phantom experiment, the difference between the theoretical intervention depth and the real-time image measurements was less than 1 mm (Fig. 6h). In the porcine-brain phantom experiment, the difference between the theoretical intervention depth and the measurements was approximately 1 mm (Fig. 6i). The reconstruction time of 370 ms using LSFP-Net for single group images (5 frames) is less than the acquisition time of 400 ms/group, which means that the latency time was 400 ms. Compared to other DL-based methods, LSFP-Net and L + S-Net have better generalizability (Fig. 6k). However, the computation time of LSFP-Net is less than a fraction of that L + S-Net, making it more suitable for real-time i-MRI (Fig. 6j).

We also performed 3D imaging experiments with fruit and porcine-brain phantoms (Supplementary Figs. 6 and 7). 3D stack-of-stars radial k-space data were acquired with a temporal resolution of 732.8 ms/volume (3.66 s/group). One group of volumes was reconstructed by LSFP-Net slice-by-slice, which took 2.96 s. Therefore, the latency of the real-time 3D imaging is 3.66 s.

**Cadaver head intervention**

To demonstrate the potential of the proposed i-MRI system for clinical potential applications, a cadaver head experiment was carried out. An MR-compatible camera was placed next to the MR head coil to monitor the movement of the needle (Fig. 7a, b). An MR-visible gelatin-filled glass fiber tube was used for the trajectory planning (Fig. 7c). The cerebral ventricle was selected as the target for ventricular drainage. After the intervention, the interventional position of the ceramic needle was validated by a 3D whole-brain T1-Weighted (T1W) MR scan (Fig. 7d). Finally, the needle was withdrawn (Fig. 7e). Two fully sampled 3D images (8 slices/volume) before and after the intervention showed the initial and final positions of the ceramic needle (Fig. 7f, h). As shown in Fig. 7g, undersampled 3D radial k-space data were acquired with a temporal resolution of 732.8 ms/volume (3.66 s/group). The reconstruction time for one volume was 2.96 s using LSFP-Net. Therefore, the latency of real-time imaging was 3.66 s. The interventional procedure can be visualized in the 6th slice of the volume (Fig. 7g and Supplementary Movie 6). The needle was partially visible in the 7th slice as the needle approached the lateral ventricle.

## Discussion

In this study, LSFP-Net was proposed to real-time monitor the interventional process in real time and track the position of the interventional feature for neurosurgery. By unrolling an iterative algorithm (LSFP) into a neural network, LSFP-Net showed superior performance and generalizability compared to other DL-based methods. It also required fewer iterations in the inference stage than CS-based iterative methods. The results demonstrated the promise of LSFP-Net for real-time i-MRI.

It is worth noting that the low-rank method has been established as a classical way for MR image reconstruction[19-26] and has been adopted in various scenarios[27,28]. With the group-based acquisition scheme using radial sampling, LSFP fully explores the low-rankness and sparsity of the intervention process. In addition, to avoid subproblems, a Primal Dual Fixed Point (PDFP) algorithm was used for optimization. The novelty of our method is that the LSFP-Net is specially designed for targeted intervention by leveraging the low-rankness of the background information and the sparsity of the intervention feature. This makes it especially useful for real-time interventional guidance. In addition, LSFP-Net is based on multi-coil radial sampling for online reconstruction. Moreover, LSFP-Net inherits the advantages of the LSFP model, i.e., exploiting the spatial sparsity of both low-rank and sparse components. Although unrolled networks exploiting the low-rank and sparse priors have been used for fast MRI reconstruction, such as SLR-Net[48] and L + S-Net[49], LSFP-Net differs from them in many ways. For the application scenarios, SLR-Net and L + S-Net were proposed for dynamic MRI, which were designed for offline reconstruction after all data acquisition is completed. LSFP-Net, on the other hand, was designed for real-time i-MRI. By using multi-coil golden-angle radial sampling and a group-based reconstruction, LSFP-Net meets the requirements for online reconstruction. In contrast,

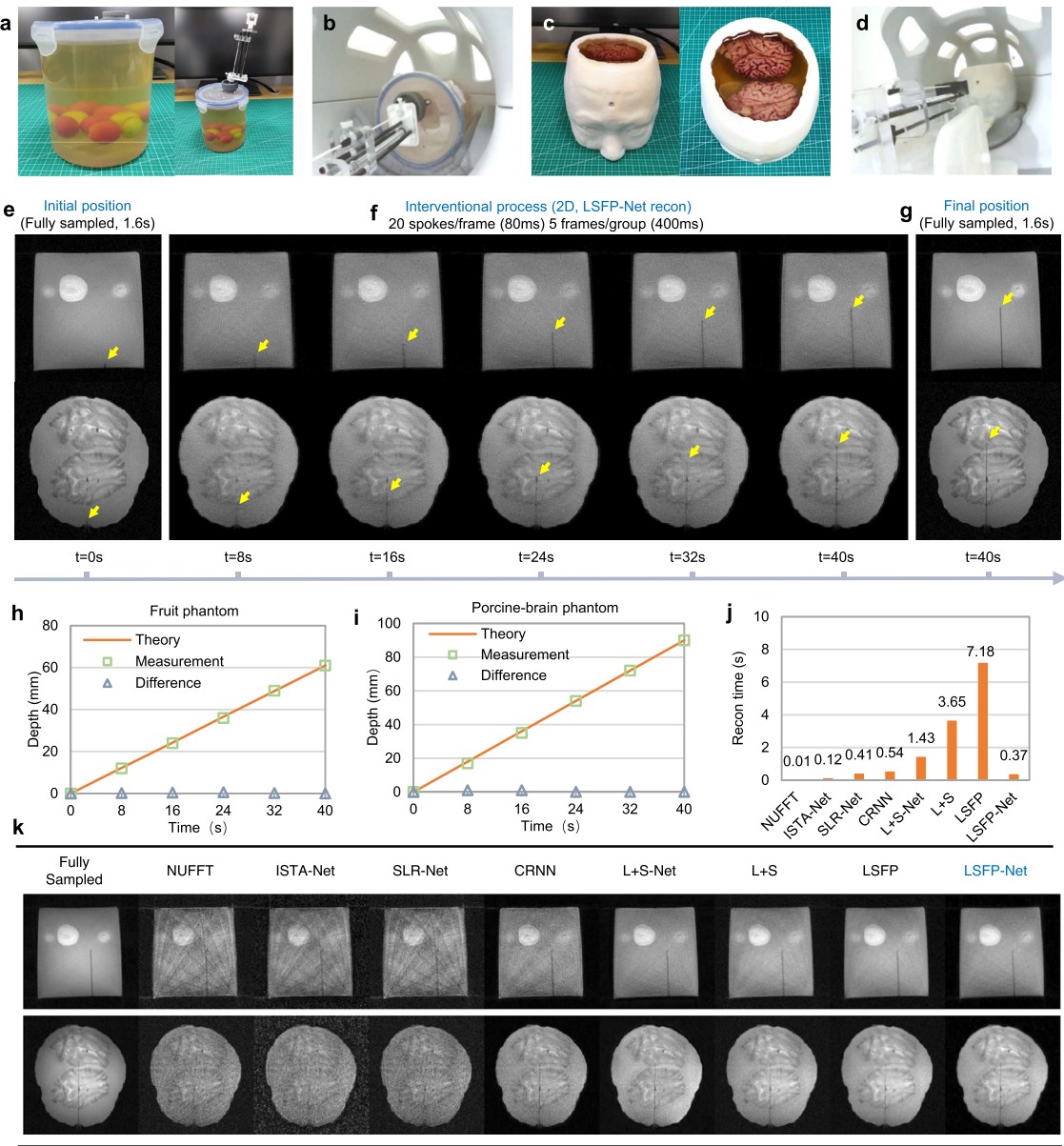

**Fig. 6 | Results of the interventional experiments with a fruit phantom and a porcine-brain phantom. a** In the fruit phantom, several red cherry tomatoes and green grapes were embedded into a cylinder filled with gelatin. **b** The interventional device was fixed on the cover of the cylinder and the combination was placed in the MR head coil. **c** Two porcine brains were embedded into a 3D-printed human skull model. **d** The porcine-brain phantom was placed in the MR head coil, and the interventional device was prepared. **e** Fully sampled 2D MR images were acquired before the intervention. **f** Real-time 2D MRI monitored the intervention using LSFP-Net for reconstruction. **g** Fully sampled 2D MR images after the intervention. **h**, **i** Comparison of the theoretical interventional depth and the measurements from real-time images. **j** A comparison of the reconstruction time of different methods on the fruit phantom experiment. **k** A comparison of different methods on two phantom intervention experiments. Source data of **h**–**j** are provided as a Source Data file.

SLR-Net and L + S-Net used Cartesian sampling and retrospective reconstruction. In terms of network structure, SLR-Net and L + S-Net only utilize the sparse prior of the sparse component and the low-rank prior of the low-rank component, but LSFP-Net exploits the spatial sparsity of both the low-rank and sparse components. LSFP-Net has also been applied to 3D imaging, expanding its potential clinical application scenarios.

Compared to Cartesian sampling, non-Cartesian sampling such as radial sampling is less sensitive to motion[9]. Many DL-based methods have been proposed for fast MRI with radial sampling[37,51–56]. Typically, continuous golden-angle radial sampling has been carried out without overlapping of radial spokes[31]. Here, by repeating the radial sampling trajectory with a period of one group of spokes, reconstruction was accelerated without the need to compute new sampling trajectories.

Different from frame-by-frame or retrospective reconstruction[16–18,29,41], the group-based reconstruction scheme[31] fully utilizes the low-rank and temporal sparsity constraints in the LSFP model and LSFP-Net. The overlapped spokes sampled between different groups were shared and used for the reconstruction of each group. Therefore, the sampling time for each frame is equivalent to the non-group-based acquisition.

Although the computation cost of SVD decomposition is high, the number of iterations and the group-based reconstruction scheme also affect the computation time. In the comparative experiments of the simulated dataset and the DBS dataset, SLR-Net with 4 iterations, L + S-Net with 15 iterations, and LSFP-Net with 2 iterations, and a group-based reconstruction scheme (5 frames per group) were used. For ISTA-Net, 12 iterations with a frame-by-frame reconstruction scheme was used. In the simulated dataset of brain intervention, the

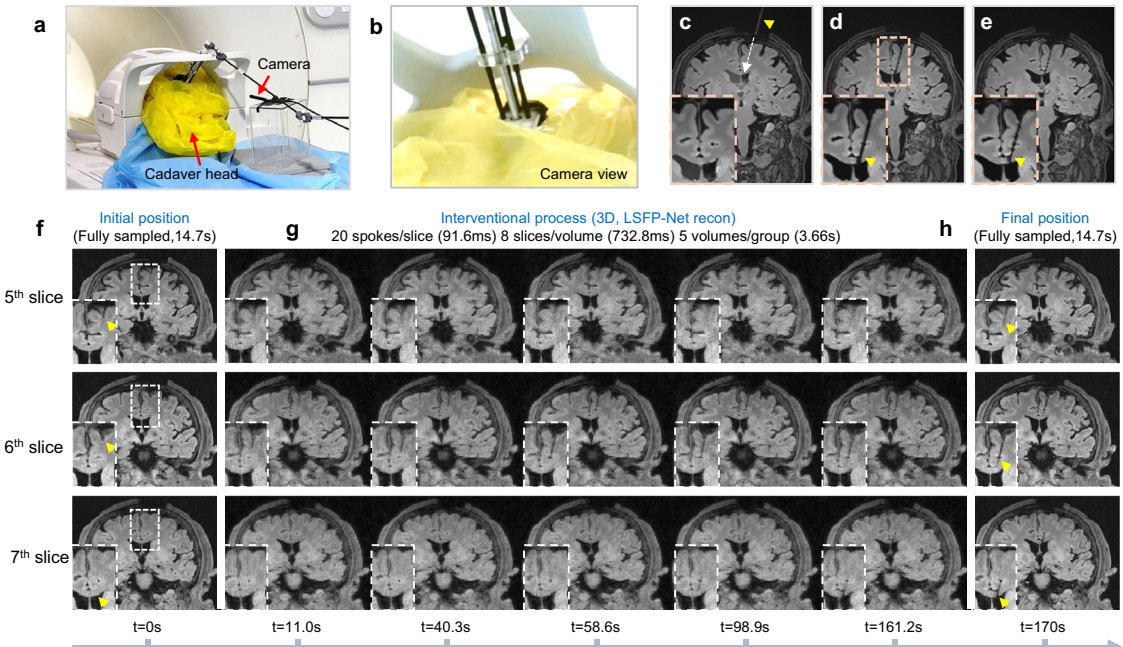

**Fig. 7 | Results of the cadaver head interventional experiment. a** The interventional device was fixed to the cadaver head. An MR-compatible camera was used to monitor the movement of the interventional device. **b** The view from the MR-compatible camera. **c** A gelatin-filled glass fiber tube (indicated by a yellow triangle) was used for the trajectory planning. **d** Whole-brain T1W MR scan after intervention shows the final position of the ceramic needle (indicated by a yellow triangle). **e** Whole-brain T1W MR scan after needle withdrawal. **f** Fully sampled MR images with 3D GRE radial sequence showing the initial needle position before the intervention. **g** Real-time MRI monitors the interventional procedure with a temporal resolution of 732.8 ms/volume (3.66 s/group). **h** Fully sampled MR images with 3D GRE radial sequence indicate the final needle position after the intervention.

computation time of SLR-Net (0.20 s), L + S-Net (0.67 s), and LSFP-Net (0.21 s) is longer than that of ISTA-Net (0.12 s). These results are consistent with reported results in existing literature[48,49].

Although many active tracking methods have been used for real-time monitoring of interventional features[57–59], they provide only positional information without background tissue. The proposed real-time i-MRI system can simultaneously monitor the interventional needle and brain tissue with high spatiotemporal resolution. This allows the operators to remotely control the intervention with real-time images, which is critical for accurate brain intervention.

Thus far, several robotic systems have been proposed for MRI-guided brain intervention[60–62]. For most of the systems, the intervention and imaging were performed separately. As a result, patients needed to be moved in and out of the scanner, compromising the accuracy and efficiency of the surgery[7]. Simultaneous intervention and imaging may be an ideal way for MRI-guided interventions. Li et al. demonstrated a 5-DOF piezoelectrically-actuated robotic system for MRI-guided DBS lead placement, allowing the intervention within the bore with synchronous imaging at 700 ms temporal resolution[63]. Unterberg-Buchwald et al. achieved real-time MRI-guided endomyocardial biopsy based on radial FLASH with nonlinear inverse reconstruction (NLINV) with an in-plane spatial resolution of $2 \times 2$ mm², a temporal resolution of 42 ms, and a latency of 0.27 s[30]. Guo et al. presented a hydraulic driving robot for MRI-guided bilateral stereotactic neurosurgery with an in-plane spatial resolution of $0.98 \times 0.98$ mm² and a temporal resolution of 17.4 s[59]. Using a selectively-actuated MRI-compatible continuum robot, Cheng et al. achieved the interactive MRI-guided cadaver neurosurgery with an in-plane spatial resolution of $1.4 \times 1.4$ mm² and a temporal resolution of 1.5 s[64]. He et al. proposed LSFP for i-MRI reconstruction and achieved an in-plane spatial resolution of $1.17 \times 1.17$ mm², a temporal resolution of 60 ms, and a latency of 10.14 s[31]. In this work, a 4-DOF MR-compatible interventional device was developed for remotely controlled intervention within the MRI bore. Together with LSFP-Net, a real-time

i-MRI system was established on a 3 T MR scanner with $1 \times 1$ mm² in-plane spatial resolution, 80 ms temporal resolution, and 0.4 s latency for 2D imaging ($1 \times 1$ mm², 732.8 ms, and 3.66 s for 3D imaging). The agreement between the theoretical intervention and the measurements from the real-time images demonstrated the real-time performance of the system. The cadaver head intervention further demonstrated the potential of the system for clinical applications.

In this study, we proposed LSFP-Net for real-time i-MRI-guided neurosurgery. For clinical application, a total of 2400 coronal slices (5 frames for each slice) generated from 23 patients with deep brain stimulation surgery were used for model training and validation. In addition, to demonstrate the performance and applicability in clinical scenarios, a custom-designed, MRI-compatible interventional device was used to construct an experimental system for brain intervention. A cadaver head was placed in a diagnostic scanner to further validate the performance of the model in neuro-intervention. With temporal resolutions of 80/732.8 ms for 2D/3D real-time MRI and in-plane spatial resolution of $1 \times 1$ mm², the proposed method showed the potential to be integrated into diagnostic scanners for image-guided neurosurgery. However, for application in live patients, a fully fledged robotic system with required regulatory and ethical approval is required to ensure the safety of the patients. Future works include training and testing with a larger sample size, and integration of the proposed model to a robotic system for additional validations.

## Methods

This study was approved by the Science and Technology Ethics Committees of Shanghai Jiao Tong University, and the Ruijin Hospital Ethics Committee of Shanghai Jiao Tong University School of Medicine.

### LSFP-Net for real-time i-MRI reconstruction

For imaging during the intervention process, the background remains relatively stable, with changes primarily centered around the target region. Therefore, the low-rankness of the background information is

leveraged by using a low-rank matrix **L**, and the sparsity of the intervention feature is incorporated through a sparse matrix **S**. The image sequence **x** could therefore be decomposed into a low-rank matrix **L** and a sparse matrix **S**, i.e., **x** = **L** + **S**. The model of LSFP was formulated as:

$$\{\mathbf{L}, \mathbf{S}\} = \arg\min_{\mathbf{L},\mathbf{S}} \frac{1}{2}\|\mathbf{E}(\mathbf{L}+\mathbf{S}) - \mathbf{d}\|_2^2 + \lambda_L\|\mathbf{L}\|_* + \lambda_s\|\nabla_t\mathbf{S}\|_1 + \lambda_L^\psi\|\psi\mathbf{L}\|_1 + \lambda_S^\psi\|\psi\mathbf{S}\|_1,$$

(1)

where $\mathbf{E} = \mathbf{\Omega}\mathbf{F}\mathbf{C}$ is the encoding operator, **C** is the coil sensitivity maps, **F** is a Fourier transform, $\mathbf{\Omega}$ is the sampling scheme. **d** is the acquired k-space data. $\nabla_t$ represents a total variation along the temporal direction of **S**. $\psi$ is the framelet transform. $\lambda_L$, $\lambda_s$, $\lambda_L^\psi$ and $\lambda_S^\psi$ are the regularization parameters. The updating steps of LSFP were summarized in Supplementary Fig. 1a.

LSFP was unrolled into a deep neural network called LSFP-Net. In LSFP-Net, the sparsifying transform $\psi$ was learnable and replaced by a combination of a 3D convolutional neural network. To improve the performance of the network, the transform pairs $\{\psi_L, \psi_L^T\}$ and $\{\psi_S, \psi_S^T\}$ are learned by different networks. The complex inputs of the convolution block were divided into real and imaginary channels. The first layer of the convolution block $\psi$ and the last layer of the convolution block $\psi^T$ have 2 convolution kernels, and the other layers have 32 convolution kernels. The size of each convolution kernel was $3 \times 3 \times 3$. Rectifier linear units (ReLU) were selected as the nonlinear activation functions.

## Simulated dataset of brain intervention

To evaluate the performance of LSFP-Net, a set of brain interventional images was simulated for training and testing. The fully sampled brain MR images from 10 healthy subjects (age 25.87 ± 2.78 years old, 5 Females, 5 Males) were collected on a 3 T MRI scanner (uMR 790, United Imaging Healthcare, Shanghai, China). All subjects provided informed consent, as approved by the Science and Technology Ethics Committees of Shanghai Jiao Tong University. For each subject, 8 coronal slices were acquired with a matrix size of 128 × 128 and 11 channels (T1W Fast Spin Echo FLAIR sequence, TR/TE = 2443/10.18 ms, flip angle = 135°, number of excitations = 1, matrix size = 128 × 128, field of view = 224 × 224 × 32 mm³, slice thickness = 4 mm). Four different interventional setups (2 unilateral and 2 bilateral) were simulated with 200 frames for each slice (Fig. 2a and Supplementary Movie 2). The training data consisted of 256 image sequences from 8 subjects. The test data set consisted of 64 image sequences from another 2 subjects. The multi-coil Non-Uniform Fast Fourier Transform (NUFFT) was adopted for simulating the golden-angle radial k-space data acquisition. For radial sampling, the number of fully sampled radial spokes is $128 \times \frac{\pi}{2} \approx 201$. The acceleration factor was $R = \text{SPF}/201$ where SPF is spokes per frame.

## Dataset from DBS patients

Postoperative MR images of patients after DBS were used to evaluate the performance of LSFP-Net (Fig. 4a). The images were acquired on a GE Healthcare Signa HDx 1.5 T MRI scanner. Three-dimensional, T1W, Fast SPoiled Gradient Recalled echo (3D-T1FSPGR) images were acquired (TR/TE = 8.23/2.6 ms, flip angle = 20°, number of excitations = 1, matrix size = 512 × 512, field of view = 224 × 224 × 304 mm³, slice thickness = 2 mm). All patients provided informed consent, as approved by the Ruijin Hospital Ethics Committee of Shanghai Jiao Tong University School of Medicine. The interventional features (DBS electrode) were first extracted from each slice of MR images. Then, sequential interventional images were generated to simulate the procedure of the intervention. A total of 2400 coronal slices (5 frames for each slice) were generated from 23 patients for training and validation. A total of 188 slices (5 frames for each slice) from another 6 patients

were used for testing. The multi-coil golden-angle radial sampling strategy was used for k-space sampling. Eight sensitivity maps were simulated using a toolbox from torchkbnufft[65]. For each time frame, a total of 402 radial sampling spokes were collected with 512 readout points for each spoke.

## Real-time brain i-MRI system

In this work, the proposed LSFP-Net was integrated into a 3 T diagnostic scanner (uMR790, United Imaging, Shanghai, China) using Gadgetron[66]. Briefly, the workflow of using Gadgetron for reconstruction is: (1) k-space data acquired from an MRI scanner was transmitted to a Gadgetron sever; (2) Images were reconstructed with trained LSFP-Net in Gadgetron server and transmitted to MRI host computer; (3) the reconstructed images were displayed on the MRI console. In this way, LSFP-Net can be readily integrated into any MRI scanners that support the Gadgetron framework. During the intervention, once the radial k-space data were acquired, they were transmitted to a Gadgetron server installed with an Ubuntu 20.04 LTS (64-bit) operating system and equipped with an AMD ryzen 9 5950× central processing unit (CPU) and NVIDIA RTX 3090 graphics processing unit (GPU, 24 GB memory). The reconstructed images were then displayed online on the MR console.

The custom-built MR-compatible interventional device has 4 degrees of freedom (DOFs). A 3-DOF ball-joint mechanism (3D-printed epoxy, ± 30°) was used for manual trajectory adjustment. A 1-DOF lead screw-nut mechanism (PMMA, aluminum, and carbon-fiber tubes, 88 mm) was remotely actuated by a stepper motor. The dimensions of the device are about 50 × 60 × 170 mm³ and the weight is ~60 g. The screw rod and the motor were connected by torque rods and a stainless-steel flexible shaft. The torque rod consisted of several universal joints (3D-printed nylon) and carbon-fiber tubes ($\phi$ = 4 mm). All components in the MR scanner room, including the interventional device and torque rod, are MR-compatible. The ferromagnetic parts were all in the control room.

## Phantom experiments

A 2D gradient echo (GRE) sequence with golden-angle radial sampling was used with the following sequence parameters: FOV = 256 × 256 mm², acquisition matrix = 256 × 256, slice thickness = 5 mm, channels = 17, TR/TE = 4/2.01 ms, and flip angle = 30°. A total of 100 group data (100 × 100 spokes) were acquired with an acquisition time of 40 seconds. Two fully sampled images were acquired before and after the intervention for comparison. For radial sampling, the number of fully sampled radial spokes is $256 \times \frac{\pi}{2} \approx 402$ with an acquisition time of 1.6 s. For the 3D case, a 3D GRE sequence with a stack-of-stars golden-angle radial sampling was used with the following sequence parameters: FOV = 256 × 256 × 24 mm³, acquisition matrix = 256 × 256 × 8, slice thickness = 3 mm, channels = 17, TR/TE = 4.58/1.76 ms, and flip angle = 10°. As in the 2D case, 20 radial spokes per slice and 5 volumes per group were used for reconstruction. A total of 11 group data were acquired with an acquisition time of 40.3 s.

## Cadaver head experiment

The cadaver study was performed with approval from the Ruijin Hospital Ethics Committee of Shanghai Jiao Tong University School of Medicine. Anatomical brain images were acquired using a T1W, Fast SPoiled GRadient Echo (3D T1 FSP GRE) sequence. The following parameters were used: TR/TE = 7.22/3.1 ms, flip angle = 8°, number of excitations = 1, matrix size = 300 × 320 × 208, field of view = 240 × 256 × 166.4 mm³, and slice thickness = 0.8 mm. These parameters resulted in a voxel volume of 0.8 mm³. A 3D GRE sequence with a stack-of-the-stars golden-angle radial sampling was used with the same sequence parameters in 3D phantom experiments. Lateral ventricle intervention with a ceramic needle was imaged in real-time with the continuous acquisition of the 3D radial k-space.

## Comparison with other algorithms

In this study, with 10 spokes for the reconstruction of each frame (acceleration factor $R = 20$), we compared the results of LSFP-Net with two iterative CS methods (L + S[23] and LSFP[31]) and four DL-based methods (CRNN[40], ISTA-Net[44], SLR-Net[48], and L + S-Net[49]). For detailed performance comparison using both the simulated and DBS datasets, CRNN was kept with its original settings as a reference. A similar number of model parameters were kept by varying unfolding iterations for the DL-based unrolled networks. Specifically, a total of 12, 4, 15, and 2 iterations were used for ISTA-Net, SLR-Net, L + S-Net, and LSFP-Net, respectively (Supplementary Table 1). Based on the comparative results (Figs. 2 and 3, and Supplementary Table 1), LSFP-Net with 3 iterations was chosen for the phantom and cadaveric experiments by considering the tradeoff between reconstruction quality and computation time.

## Performance evaluation

Peak signal-to-noise ratio (PSNR) and structural similarity index (SSIM) were used to evaluate the performance of i-MRI. The Adam optimizer[67] with parameters $\beta_1 = 0.9$, $\beta_2 = 0.999$, $\varepsilon = 10^{-8}$, and a learning rate of 0.0001, and a batch size of 1 (due to the limited GPU memory) were used for training. The models were implemented using Pytorch based on the torchKbNufft[65] with an Ubuntu 20.04 LTS (64-bit) operating system equipped with an i9-12900K CPU and NVIDIA RTX 3080 Ti GPU (12 GB memory). Training LSFP-Net took ~5 hours for 100 epochs.

## Statistics and reproducibility

The code and datasets used for training and testing the deep-learning models are made publicly available for reproducibility[68,69]. The mean values and standard deviations of PSNR/SSIM were calculated in all comparative experiments. No statistical method was used to pre-determine the sample size. No data were excluded from the analyses. In both simulated datasets of brain intervention and DBS patients, the training and testing datasets were randomly allocated.

## Reporting summary

Further information on research design is available in the Nature Portfolio Reporting Summary linked to this article.

# Data availability

The datasets for training and testing LSFP-Net have been deposited in Figshare under accession code DOI link[68]. Source data are provided with this paper.

# Code availability

The code for implementation of the LSFP-Net is made publicly available[69]. Other information is available from the corresponding author (Y.F.) upon request.

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

## Acknowledgements
Funding supports from the National Key R&D Program of China (grant 2022YFB4702704, Y.F., 2022YFB4702700, G.-Z.Y.), the National Natural Science Foundation of China (grant 32322042, Y.F., 32271359, Y.F.,

12090024, X.Z.), the Natural Science Foundation of Shanghai (grant 22ZR1429600, Y.F.), Science and Technology Commission of Shanghai Municipality (grant 20DZ2220400, G.-Z.Y.), and Shanghai Municipal Science and Technology Major Project (grant 2021SHZDZX, G.-Z.Y., 2021SHZDZX0102, X.Z.) are acknowledged. Suhao Qiu, Linghan Kong, Ruiyang Zhao, and Runke Wang are acknowledged for dataset preparation and MR sequence optimization. Yi-Xin Pan is acknowledged for the helpful discussion of the experiments. We also thank the Student Innovation Center and the Center for High Performance Computing (π 2.0 cluster) at Shanghai Jiao Tong University for providing us the computing services.

## Author contributions

Z.H., Y.-N.Z., Y.H., X.Z. and Y.F. carried out network implementation and image analysis. Z.H., Yu C. and Y.F. designed the experiment device. T.W., C.Z., B.S. and F.Y. collected the clinical image data. Z.H., Yi C., Y.S., Q.S. and Y.F. carried out the intervention experiments. Z.H., Y.-N.Z., Y.H., X.Z., G.-Z.Y. and Y.F. wrote and edited the manuscript. Y.F., G.-Z.Y. and Q.S. conceived and planned the study. All authors reviewed and commented on the manuscript.

## Competing interests

The authors declare no competing interests.
