## [Peer Review File · Nature Communications]

A deep unrolled neural network for real-time MRI-guided
brain interventionReviewer #1 (Remarks to the Author):

This paper introduces a deep neural network unrolled from the LSFP model, which employs Low-rank and Sparsity decomposition with Framelet transform and Primal dual fixed point optimization, for navigation and targeting in real-time MRI. The proposed method is integrated into a 3T MRI device and its effectiveness is validated through phantom and in vivo experiments. However, the paper seems lacking of methodological innovation and insufficient clinical case studies to establish its effectiveness.

1. The paper proposes a deep neural network unrolled from the LSFP model to achieve real-time MRI. However, the low-rank constraint in the LSFP model necessitates simultaneous computation of a group of data (multiple frames), thereby preventing true "real-time" (frame-by-frame) imaging [1-5].
2. From a methodological standpoint, the proposed LSFP model lacks innovation as similar models have already been proposed [6-7]. Moreover, the unfolding of the LSFP model into a deep neural network resembles methods like L+S Net [8]. Although the authors argue that these methods are designed for offline reconstruction and Cartesian sampling, they can be easily adapted to the scenario presented in this paper. Therefore, the authors should discuss the advantages of their proposed method and conduct comparative experiments.
3. The paper asserts that the proposed model can be readily integrated into diagnostic scanners for image-guided neurosurgery. However, the authors fail to provide adequate clinical cases or experimental data to substantiate this claim.
4. Regarding the comparative experiments, it is important to explain why methods such as CRNN perform worse than traditional CS methods. The authors need to explain the reasons.
5. Lastly, the LSFP unrolled network involves SVD decomposition, yet its reconstruction time is nearly comparable to ISTA-Net. The authors also need to explain the reasons for this.

Reference :

- [1] Jung H, Sung K, Nayak K S, et al. k-t FOCUSS: a general compressed sensing framework for high resolution dynamic MRI. *Magnetic Resonance in Medicine: An Official Journal of the International Society for Magnetic Resonance in Medicine*, 2009, 61(1): 103-116.
- [2] Uecker M, Zhang S, Voit D, et al. Real-time MRI at a resolution of 20 ms. *NMR in Biomedicine*, 2010, 23(8): 986-994.
- [3] Majumdar A, Ward R K, Aboulnasr T. Compressed sensing based real-time dynamic MRI reconstruction. *IEEE Transactions on Medical Imaging*, 2012, 31(12): 2253-2266.
- [4] Chen C, Li Y, Axel L, et al. Real time dynamic MRI with dynamic total variation, *MICCA*, 2014: 138-145.
- [5] Terpstra M L, Maspero M, d'Agata F, et al. Deep learning-based image reconstruction and motion estimation from undersampled radial k-space for real-time MRI-guided radiotherapy. *Physics in Medicine & Biology*, 2020, 65(15): 155015.
- [6] Jin K H, Ye J C. Sparse and low-rank decomposition of a Hankel structured matrix for impulse noise removal. *IEEE Transactions on Image Processing*, 2017, 27(3): 1448-1461.
- [7] Li H, He X, Tao D, et al. Joint medical image fusion, denoising and enhancement via discriminative low-rank sparse dictionaries learning. *Pattern Recognition*, 2018, 79: 130-146.
- [8] Huang W, Ke Z, Cui Z X, et al. Deep low-rank plus sparse network for dynamic MR imaging. *Medical Image Analysis*, 2021, 73: 102190.

Reviewer #2 (Remarks to the Author):

This paper presents a method for real-time MRI-guided brain intervention using an integrated

interventional MRI (i-MRI) system and a custom-designed interventional device. The method involves continuously acquiring radial k-space data during the intervention, reconstructing it online, and immediately sending it to the console for visualization. The results show that the system achieves high temporal resolutions and low latencies, allowing for accurate and timely guidance during interventions.

Although the method of this paper, Low rank and Sparsity decomposition with Framelet and Primal dual fixed point unrolled into a deep neural network (LSFP-Net), is based on the previously proposed Low-rank and Sparsity decomposition with Framelet (LSF) model for i-MRI reconstruction, the paper introduces a new implementation and integration of LSFP-Net with an interventional MRI system and a custom-designed interventional device, which is a novel contribution.

Contribution:

This paper provides a potential solution to tackle the challenge of balancing temporal and spatial resolution for real-time interventional MRI by unrolling the iterative LSFP algorithm into a neural network. The performance of the system was evaluated using phantom and cadaver studies. The results of phantom experiments show that the 2D/3D real-time MRI was achieved with temporal resolutions of 80/732.8 ms, latencies of 0.4/3.66 s, including data communication, processing and reconstruction time, and in-plane spatial resolution of 1x1 mm², which demonstrate the real-time imaging and remotely actuated intervention capabilities of the system.

Additional Comments:

1. There are other literatures for real-time MRI, authors should include more comparisons with the state-of-the-art work in the introduction section. It is unclear how the proposed method compares to other available options in terms of accuracy, speed, and ease of use.
2. The proposed method was only tested on a limited number of clinical cases, and further studies are needed to validate its effectiveness in a larger sample size. The system was only evaluated using phantom and cadaver studies, which may not fully represent the complexity and variability of live brain interventions.
3. The proposed method requires a custom-designed interventional device, which may limit its widespread adoption in clinical practice.
4. While the proposed LSFP-Net achieved real-time i-MRI reconstruction with high temporal and spatial resolution, the study did not compare its performance to other existing image guidance systems.

Response to reviewers

=====

Thank you again for submitting your manuscript "A deep unrolled neural network for real-time MRI-guided brain intervention" to Nature communications. We have now received reports from 2 reviewers and, after careful consideration, we have decided to invite a major revision of the manuscript.

As you will see from the reports copied below, the reviewers raise important concerns. We find that these concerns limit the strength of the study, and therefore we ask you to address them with additional work. Without substantial revisions, we will be unlikely to send the paper back to review. In particular, the reviewers raise concerns regarding the comparison to other approaches. Moreover, we would encourage you to discuss the limitations of your work for clinical applications.

If you feel that you are able to comprehensively address the reviewers' concerns, please provide a point-by-point response to these comments along with your revision. Please show all changes in the manuscript text file with track changes or colour highlighting. If you are unable to address specific reviewer requests or find any points invalid, please explain why in the point-by-point response.

Response: We thank the editor and reviewers for their detailed and constructive comments. We have conducted additional comparative experiments to emphasize the innovation and improvement of the proposed method. The current limitations of our work are also discussed. Point-to-point responses to the questions and comments raised are given below, with the corresponding changes made to the manuscript marked in blue. We hope that our response and the revised manuscript can address your and the reviewers' concerns.

Response to Reviewers

Reviewer #1:

This paper introduces a deep neural network unrolled from the LSFP model, which employs Low-rank and Sparsity decomposition with Framelet transform and Primal dual fixed point optimization, for navigation and targeting in real-time MRI. The proposed method is integrated into a 3T MRI device and its effectiveness is validated through phantom and in vivo experiments. However, the paper seems lacking of methodological innovation and insufficient clinical case studies to establish its effectiveness.

Response: Thank you for the summary and constructive critiques. Additional comparative experiments have been conducted to emphasize the innovation and improvement of the proposed LSFP-Net compared to the current state-of-the-art. In addition, we have addressed the limitations of our work for clinical applications in the discussion section. Point-to-point responses to your questions and comments raised are given below. We hope that our response and the revised manuscript can address all your concerns.

1. *The paper proposes a deep neural network unrolled from the LSFP model to achieve real-time MRI. However, the low-rank constraint in the LSFP model necessitates simultaneous computation of a group of data (multiple frames), thereby preventing true "real-time" (frame-by-frame) imaging [1-5].*

[1] Jung H, Sung K, Nayak K S, et al. *k-t FOCUSS: a general compressed sensing framework for high resolution dynamic MRI. Magnetic Resonance in Medicine: An Official Journal of the International Society for Magnetic Resonance in Medicine*, 2009, 61(1): 103-116.

[2] Uecker M, Zhang S, Voit D, et al. *Real-time MRI at a resolution of 20 ms. NMR in Biomedicine*, 2010, 23(8): 986-994.

[3] Majumdar A, Ward R K, Aboulnasr T. *Compressed sensing based real-time dynamic MRI reconstruction. IEEE Transactions on Medical Imaging*, 2012, 31(12): 2253-2266.

[4] Chen C, Li Y, Axel L, et al. *Real time dynamic MRI with dynamic total variation, MICCA*, 2014: 138-145.

[5] Terpstra M L, Maspero M, d'Agata F, et al. *Deep learning-based image reconstruction and motion estimation from undersampled radial k-space for real-time MRI-guided radiotherapy. Physics in Medicine & Biology*, 2020, 65(15): 155015.

Response: Thank you for pointing this out. In this work, the group-based reconstruction scheme allows the low-rank and temporal sparsity constraints in the LSFP model and LSFP-Net. We agree that this strategy may affect the temporal resolution in part due to the simultaneous reconstruction of a group of data (multiple frames). However, as we discussed in our previous work [1], similar to the sliding window or data-sharing scheme, the group-based reconstruction can achieve a temporal resolution of 60 ms, satisfying the frame-by-frame imaging requirement for the relatively slow neuro-intervention (**Response Fig. 1**). To prevent redundant computation, the overlapped spokes sampled between different groups were shared and used for the reconstruction of each group. In this way, the sampling time for each frame is equivalent to the non-group-based acquisition.

The following sentences have been added to the introduction section (lines 45-58):

“Over the past decade, compressed sensing (CS)-based methods exploiting data sparsity have been used to accelerate imaging speed^{14,15}. By further utilizing the temporal information, CS-based k-t methods have been proposed for real-time dynamic MRI¹⁶⁻¹⁸. Low-rank matrix imaging involving one or more dimensions has become an established method for fast MR imaging¹⁹⁻²². Decomposing the data matrix into a low-rank component (L) and a sparse component (S), i.e., low-rank plus sparse decomposition (L+S) or robust principle component analysis (RPCA), has been proposed and applied for dynamic MRI²³⁻²⁶ and other fields^{27,28}. In addition, high temporal resolution imaging has been achieved using a nonlinear inverse reconstruction method with undersampled radial sampling for real-time cardiovascular MRI^{29,30}. However, the relatively low spatial resolution may not satisfy the requirements for neurosurgery. Recently, a combination of Low-rank and Sparsity decomposition with Framelet transform and Primal dual fixed point optimization (LSFP) was proposed for i-MRI reconstruction with a group-based reconstruction scheme³¹. However, complex parameter tuning and long computation time are required, which cannot satisfy the online reconstruction requirement of real-time i-MRI.”

For clarity, we have also added the following sentences to the discussion section (lines 288-292):

“Different from frame-by-frame or retrospective reconstruction^{16-18,29,41}, the group-based reconstruction scheme³¹ fully utilizes the low-rank and temporal sparsity constraints in the LSFP model and LSFP-Net. The overlapped spokes sampled between different groups were shared and used for the reconstruction of each group. Therefore, the sampling time for each frame is equivalent to the non-group-based acquisition.”

Fig. 1. Illustrations of the reconstruction scheme. (a) Conventional dynamic image reconstruction based on a retrospective scheme. One vertical line represents one golden-angle spoke. (b) The proposed group-based reconstruction method for real-time i-MRI reconstruction, without overlap among different groups. (c) The proposed group-based reconstruction method for real-time i-MRI reconstruction, with overlap among different groups.

Response Fig. 1. Illustrations of the reconstruction scheme. “If s is the number of radial spokes used for the reconstructing of one frame, m is the number of frames used for each reconstruction group, p is the number of overlapped frames between the adjacent groups, and TR is the acquisition time of one spoke, then the temporal resolution is $TR \times s \times (m-p)$. Fig. 1(b) and (c) show the cases of $p=0$ and $p=m-1$, respectively. In practice, $p=m-1$ will achieve the highest temporal resolution but will introduce repeated computation of the overlapped frames. Here, we adopted $p=0$ for speed consideration.” (He et al., 2022)

2. From a methodological standpoint, the proposed LSFP model lacks innovation as similar models have already been proposed [6-7]. Moreover, the unfolding of the LSFP model into a deep neural network resembles methods like L+S Net [8]. Although the authors argue that these methods are designed for offline reconstruction and Cartesian sampling, they can be easily adapted to the scenario presented in this paper. Therefore, the authors should discuss the advantages of their proposed method and conduct comparative experiments.

[6] Jin K H, Ye J C. Sparse and low-rank decomposition of a Hankel structured matrix for impulse noise removal. *IEEE Transactions on Image Processing*, 2017, 27(3): 1448-1461.

[7] Li H, He X, Tao D, et al. Joint medical image fusion, denoising and enhancement via discriminative low-rank sparse dictionaries learning. *Pattern Recognition*, 2018, 79: 130-146.

[8] Huang W, Ke Z, Cui Z X, et al. Deep low-rank plus sparse network for dynamic MR imaging. *Medical Image Analysis*, 2021, 73: 102190.

Response: The low-rank method has indeed been established for MR image reconstruction [2-9], and has been adopted in various scenarios [10, 11]. Here, with the group-based acquisition scheme of radial sampling, LSFP fully explores the low-rankness and sparsity of the intervention process. In addition, to avoid subproblems, we used a Primal-Dual Fixed Point (PDFP) algorithm for optimization. For clarity, we have added the following sentences to the discussion section (lines 265-273) to emphasize the novelty of our proposed method:

“It is worth noting that the low-rank method has been established as a classical way for MR image reconstruction ¹⁹⁻²⁶ and has been adopted in various scenarios ^{27,28}. With the group-based acquisition scheme of radial sampling, LSFP fully explores the low-rankness and sparsity of the intervention process. In addition, to avoid subproblems, a Primal-Dual Fixed Point (PDFP) algorithm was used for optimization. The novelty of our method is that the LSFP-Net is specially designed for intervention, which used the low rankness of the background information and the sparsity of the intervention feature. This makes it especially useful for imaging the intervention process. In addition, LSFP-Net is based on multi-coil radial sampling for online reconstruction. Moreover, LSFP-Net inherits the advantages of the LSFP model, i.e., exploiting the spatial sparsity of both low-rank and sparse components.”

The differences were presented in the discussion section (lines 273-283):

“ Although unrolled networks exploiting the low-rank and sparse priors have been used for fast MRI reconstruction, such as SLR-Net ⁴⁸ and L+S-Net ⁴⁹, LSFP-Net differs from them in many ways. For the application scenarios, SLR-Net and L+S-Net were proposed for dynamic MRI, which were designed for offline reconstruction after all data acquisition is completed. LSFP-Net, on the other hand, was designed for real-time i-MRI. By using multi-coil golden-angle radial sampling and a group-based reconstruction, LSFP-Net meets the requirements for online reconstruction. In contrast, SLR-Net and L+S-Net used Cartesian sampling and retrospective reconstruction. In terms of network structure, SLR-Net and L+S-Net only utilize the sparse prior of the sparse component and the low-rank prior of the low-rank component, but LSFP-Net exploits the spatial sparsity of both the low-rank and sparse components. LSFP-Net has also been applied to 3D imaging, expanding its potential clinical application scenarios.”

We also conducted detailed comparative experiments as shown in Fig. 2 (b)-(f), Fig. 3(a)-(c), Fig. 4(b)-(e), and Fig. 6(k). Results present that the performance of LSFP-Net outperformed SLR-Net and L+S-Net, demonstrating the improvement of the LSFP-Net.

Fig. 2. (b) A comparison of different methods. 10 spokes were used for the reconstruction of each frame (acceleration factor R=20). Five frames per group were used for L+S, LSFP, CRNN, SLR-Net, L+S-Net and LSFP-Net. (c) The magnified view of the interventional features. The area of the interventional feature is indicated by the red dashed box in (b). (d)-(f) The quantitative metrics of the different methods.

Fig. 3. Results of different parameters for LSFP-Net. (a)-(c) A comparison of different methods with R= 40, 25, and 10.

Fig. 4. A comparison of different methods on the simulated DBS electrode placement dataset. (a) The dataset preparation is based on the postoperative MRI of DBS electrode placement. **(b)** The reconstruction results from different methods using 10 spokes. **(c-d)** The quantitative metrics of different methods.

Fig. 6. Results of the interventional experiments with a fruit phantom and a porcine-brain phantom. (k) A comparison of different methods on two phantom intervention experiments.

3. The paper asserts that the proposed model can be readily integrated into diagnostic scanners for image-guided neurosurgery. However, the authors fail to provide adequate clinical cases or experimental data to substantiate this claim.

Response: In this study, we proposed LSFP-Net for real-time MRI-guided neurosurgery. For clinical application, a total of 2400 coronal slices (5 frames for each slice) generated from 23 patients with deep brain stimulation surgery were used for model training and validation. In addition, to demonstrate the performance and applicability in clinical scenarios, a custom-designed, MRI-compatible interventional device was used to construct an experimental system for brain intervention. A cadaver head was placed in a diagnostic scanner to further validate the performance of the model in neuro-intervention. The real-time imaging for the cadaver intervention showed the proposed model could be used for guiding neuro-intervention well. For patient studies, a fully fledged robotic system with ethical regulatory approval is required to ensure the safety of the patients. The development of the robotic system is underway in our institution but is beyond the scope of this study. We believe the training data from patients, phantom, and cadaver head experiments can sufficiently demonstrate the performance of the proposed model. We have modified the following sentences in the abstract and discussion to clarify this fact:

“The results demonstrated that the proposed method enables real-time monitoring of the remote-controlled brain intervention, and showed the potential to be integrated into diagnostic scanners for image-guided neurosurgery.” (line 31, last sentence of the abstract)

“In this study, we proposed LSFP-Net for real-time i-MRI guided neurosurgery. For clinical application, a total of 2400 coronal slices (5 frames for each slice) generated from 23 patients with deep brain stimulation surgery were used for model training and validation. In addition, to demonstrate the performance and applicability in clinical scenarios, a custom-designed, MRI-compatible interventional device was used to construct an experimental system for brain intervention. A cadaver head was placed in a diagnostic scanner to further validate the performance of the model in neuro-intervention. With temporal resolutions of 80/732.8 ms for 2D/3D real-time MRI and in-plane spatial resolution of 1x1 mm², the proposed method showed the potential to be integrated into diagnostic scanners for image-guided neurosurgery. However, for application in live patients, a fully fledged robotic system with required regulatory and ethical approval is required to ensure the safety of the patients. Future works include training and testing with a larger sample size, and integration of the proposed model to a robotic system for additional validations.” (lines 325-336, last paragraph of discussion)

In addition, we also added a few details of the deployment of LSFP-Net to the clinical scanner in the methods section (lines 381-386):

“In this work, the proposed LSFP-Net was integrated into a 3T diagnostic scanner (uMR790, United Imaging, Shanghai, China) using Gadgetron⁶⁷. Briefly, the workflow of using Gadgetron for reconstruction is: 1) k-space data acquired from an MRI scanner is transmitted to a Gadgetron sever; 2) Images were reconstructed with trained LSFP-Net in Gadgetron server and transmitted to MRI host computer; 3) the reconstructed images were displayed on the MRI console. In this way, LSFP-Net can be readily integrated into any MRI scanners that support the Gadgetron framework.”

4. *Regarding the comparative experiments, it is important to explain why methods such as CRNN perform worse than traditional CS methods. The authors need to explain the reasons.*

Response: We apologize for the confusion. The results of the comparative experiments were double-checked and updated, In simulated and DBS datasets, the performance of the DL-based

methods (LSFP-Net, L+S-Net, SLR-Net, CRNN, and ISTA-Net) are better than that of the CS methods (L+S and LSFP), as shown in Fig. 2 (b)-(f), and Fig. 4(b)-(e).

5. Lastly, the LSFP unrolled network involves SVD decomposition, yet its reconstruction time is nearly comparable to ISTA-Net. The authors also need to explain the reasons for this.

Response: Thanks for pointing this out. The following explanations were added to the discussion section (lines 293-298):

“Although the computation cost of SVD decomposition is high, the number of iterations and the group-based reconstruction scheme also affect the computation time. For SLR-Net, L+S-Net, and LSFP-Net, 3 iterations and a group-based reconstruction scheme (5 frames per group) were used. For ISTA-Net, 9 iterations with a frame-by-frame reconstruction scheme was used. In the simulated dataset of brain intervention, the computation time of SLR-Net (0.16s), L+S-Net (0.18s), and LSFP-Net (0.29s) is longer than that of ISTA-Net (0.09s). These results are consistent with reported results in existing literature ^{48,49}”.

Reviewer #2:

This paper presents a method for real-time MRI-guided brain intervention using an integrated interventional MRI (i-MRI) system and a custom-designed interventional device. The method involves continuously acquiring radial k-space data during the intervention, reconstructing it online, and immediately sending it to the console for visualization. The results show that the system achieves high temporal resolutions and low latencies, allowing for accurate and timely guidance during interventions. Although the method of this paper, Low rank and Sparsity decomposition with Framelet and Primal dual fixed point unrolled into a deep neural network (LSFP-Net), is based on the previously proposed Low-rank and Sparsity decomposition with Framelet (LSF) model for i-MRI reconstruction, the paper introduces a new implementation and integration of LSFP-Net with an interventional MRI system and a custom-designed interventional device, which is a novel contribution.

Contribution:

This paper provides a potential solution to tackle the challenge of balancing temporal and spatial resolution for real-time interventional MRI by unrolling the iterative LSFP algorithm into a neural network. The performance of the system was evaluated using phantom and cadaver studies. The results of phantom experiments show that the 2D/3D real-time MRI was achieved with temporal resolutions of 80/732.8 ms, latencies of 0.4/3.66 s, including data communication, processing and reconstruction time, and in-plane spatial resolution of 1x1 mm², which demonstrate the real-time imaging and remotely actuated intervention capabilities of the system.

Response: Thank you for recognizing our efforts in real-time i-MRI guided brain interventions. We have carefully gone through all your comments, which have further helped us to improve the manuscript. We hope that our response and the revised manuscript can address all your concerns.

Additional comments:

1. *There are other literatures for real-time MRI, authors should include more comparisons with the state-of-the-art work in the introduction section. It is unclear how the proposed method compares to other available options in terms of accuracy, speed, and ease of use.*

Response: Thank you. We have added more references for real-time MRI in the introduction section. Additional comparisons were made with the state-of-the-art algorithms. The second, third, and fourth paragraphs of the introduction section have been rewritten as flows (lines 45-79):

“ ...

Over the past decade, compressed sensing (CS)-based methods exploiting data sparsity have been used to accelerate imaging speed^{14,15}. By further utilizing the temporal information, CS-based k-t methods have been proposed for real-time dynamic MRI¹⁶⁻¹⁸. Low-rank matrix imaging involving one or more dimensions has become an established method for fast MR imaging¹⁹⁻²². Decomposing the data matrix into a low-rank component (L) and a sparse component (S), i.e., low-rank plus sparse decomposition (L+S) or robust principle component analysis (RPCA), has been proposed and applied for dynamic MRI²³⁻²⁶ and other fields^{27,28}. In addition, high temporal resolution imaging has been achieved using a nonlinear inverse reconstruction method with undersampled radial sampling for real-time cardiovascular MRI^{29,30}. However, the relatively low spatial resolution may not satisfy

the requirements for neurosurgery. Recently, a combination of Low-rank and Sparsity decomposition with Framelet transform and Primal dual fixed point optimization (LSFP) was proposed for i-MRI reconstruction with a group-based reconstruction scheme³¹. However, complex parameter tuning and long computation time are required, which cannot satisfy the online reconstruction requirement of real-time i-MRI.

Deep learning (DL) can greatly improve reconstruction quality and accelerate computation speed, making it especially useful in fast MRI³²⁻³⁴. Typical DL networks include AUTOMAP³⁵, GAN³⁶, U-nets³⁷, transformers³⁸, and diffusion models³⁹. For further utilizing temporal information, a convolutional recurrent neural network (CRNN) was proposed for dynamic MRI⁴⁰. Similarly, Jaubert et al. developed a deep artifact suppression method using recurrent U-Nets for real-time cardiac MRI³⁷. A DL-based image reconstruction and motion estimation from undersampled radial k-space has also been applied to real-time MRI-guided radiotherapy⁴¹. However, these data-driven networks rely on large-scale training datasets and have limited interpretability and generalizability⁴². To overcome these limitations, unrolled networks were proposed. Typical models include cascaded networks⁴³, ISTA-Net (unrolling of the iterative shrinkage-thresholding algorithm)⁴⁴, ADMM-Net (unrolling of alternating direction method of multipliers method)⁴⁵, and variational network (unrolling of gradient descent algorithm)⁴⁶. An unrolled variational network with an undersampled spiral k-space trajectory was also developed for real-time cardiac MRI reconstruction⁴⁷. However, only exploiting sparse prior limits the performance of these networks. By utilizing both low-rank and sparse priors, SLR-Net⁴⁸ and L+S-Net⁴⁹ have become two state-of-the-art unrolled networks for dynamic MRI reconstruction. However, SLR-Net and L+S-Net are designed for retrospective reconstruction with Cartesian sampling, which cannot satisfy the online reconstruction requirement of real-time i-MRI.

In this study, we proposed LSFP-Net for i-MRI reconstruction by unrolling the iterative LSFP algorithm into a neural network. The low-rank and sparse priors and spatial sparsity of both low-rank and sparse components are utilized. The group-based reconstruction with periodic radial sampling makes LSFP-Net satisfy the online reconstruction requirement for real-time i-MRI.

...”

2. The proposed method was only tested on a limited number of clinical cases, and further studies are needed to validate its effectiveness in a larger sample size. The system was only evaluated using phantom and cadaver studies, which may not fully represent the complexity and variability of live brain interventions.

Response: Thank you. In this study, we proposed LSFP-Net for real-time MRI-guided neurosurgery. For clinical application, a total of 2400 coronal slices (5 frames for each slice) generated from 23 patients with deep brain stimulation surgery were used for model training and validation. In addition, to demonstrate the performance and applicability in clinical scenarios, a custom-designed, MRI-compatible interventional device was used to construct an experimental system for brain intervention. A cadaver head was placed in a diagnostic scanner to further validate the performance of the model in neuro-intervention. The real-time imaging of the brain intervention showed the proposed model could be used for guiding neuro-intervention well. However, for application in live patients, a fully fledged robotic system with regulatory and ethical approval is required to ensure the safety of the patients. The development of the robotic system is underway in our institution but is beyond the scope of this study. Therefore, we believe the training data from patients, phantom, and cadaver

head experiments can demonstrate the performance of the proposed model. Future work will integrate the proposed model with the robotic system for clinical applications. The limitations of this work were added to the discussion section (lines 325-336, the last paragraph of the discussion):

“In this study, we proposed LSFP-Net for real-time i-MRI guided neurosurgery. For clinical application, a total of 2400 coronal slices (5 frames for each slice) generated from 23 patients with deep brain stimulation surgery were used for model training and validation. In addition, to demonstrate the performance and applicability in clinical scenarios, a custom-designed, MRI-compatible interventional device was used to construct an experimental system for brain intervention. A cadaver head was placed in a diagnostic scanner to further validate the performance of the model in neuro-intervention. With temporal resolutions of 80/732.8 ms for 2D/3D real-time MRI and in-plane spatial resolution of 1x1 mm², the proposed method showed the potential to be integrated into diagnostic scanners for image-guided neurosurgery. However, for application in live patients, a fully fledged robotic system with required regulatory and ethical approval is required to ensure the safety of the patients. Future works include training and testing with a larger sample size, and integration of the proposed model to a robotic system for additional validations.”

3. *The proposed method requires a custom-designed interventional device, which may limit its widespread adoption in clinical practice.*

Response: Thank you. The focus of this study is to develop and validate the imaging scheme and reconstruction algorithm for real-time i-MRI. The custom-designed interventional device was only intended for experimental purposes, providing repeatable, remote-controlled intervention for the ex vivo studies. For interventions of live patients, a fully fledged robotic system needs to be used to ensure the safety of the patients. The development of the robotic system is underway in our institution but is beyond the scope of this study, as it also needs to go through the lengthy regulatory and ethical approval processes.

4. *While the proposed LSFP-Net achieved real-time i-MRI reconstruction with high temporal and spatial resolution, the study did not compare its performance to other existing image guidance systems.*

Response: Thank you. As suggested, comparisons between the proposed method and other real-time MRI-guided intervention systems are summarized in Supplementary Table 1.

Supplementary Table 1. The comparisons of different real-time i-MRI systems

Study	Intervention	Subject	MR scanner	2D/3D imaging	Spatial resolution (mm ²)	Temporal resolution (ms)	Latency (s)
Li (2015) ²	Brain	Phantom	3T	2D	N/A	700	N/A
Unterberg-Buchwald (2017) ³	Cardiac	Animal (pig)	3T	2D	2x2	42	0.27

Guo (2018) 4	Brain	Phantom	1.5T	2D	0.98x0.98	17400	N/A
Cheng (2021) ⁵	Brain	Cadaver	3T	2D	1.4x1.4	1500	N/A
He (2022) ¹	Brain	Phantom	3T	2D	1.17x1.17	60	10.14
Ours	Brain	Cadaver	3T	2D/3D	1x1	80/732.8	0.4/3.66

We also added the following paragraph to the discussion section (lines 307-317):

“...Li et al. demonstrated a 5-DOF piezoelectrically-actuated robotic system for MRI-guided DBS lead placement, allowing the intervention within the bore with synchronous imaging at 700 ms temporal resolution⁶³. Unterberg-Buchwald et al. achieved real-time MRI-guided endomyocardial biopsy based on radial FLASH with nonlinear inverse reconstruction (NLINV) with an in-plane spatial resolution of 2x2 mm², a temporal resolution of 42 ms, and a latency of 0.27 s³⁰. Guo et al. presented a hydraulic driving robot for MRI-guided bilateral stereotactic neurosurgery with an in-plane spatial resolution of 0.98x0.98 mm² and a temporal resolution of 17.4 s⁶⁴. Using a selectively-actuated MRI-compatible continuum robot, Cheng et al. achieved the interactive MRI-guided cadaver neurosurgery with an in-plane spatial resolution of 1.4x1.4 mm² and a temporal resolution of 1.5 s⁶⁵. He et al. proposed LSFP for i-MRI reconstruction and achieved an in-plane spatial resolution of 1.17x1.17 mm², a temporal resolution of 60 ms, and a latency of 10.14 s³¹. In this work, a 4-DOF MR-compatible interventional device was developed for remotely controlled intervention within the MRI bore. Together with LSFP-Net, a real-time i-MRI system was established on a 3T MR scanner with 1x1 mm² in-plane spatial resolution, 80 ms temporal resolution, and 0.4 s latency for 2D imaging (1x1 mm², 732.8 ms, and 3.66 s for 3D imaging).”

References

- [1] Z. He *et al.*, "Low-Rank and Framelet Based Sparsity Decomposition for Interventional MRI Reconstruction," *IEEE transactions on bio-medical engineering*, vol. 69, no. 7, pp. 2294-2304, 2022-Jan-11 2022, doi: 10.1109/tbme.2022.3142129.
- [2] Z. P. Liang, "Spatiotemporal imaging with partially separable functions," (in English), *2007 4th IEEE International Symposium on Biomedical Imaging : Macro to Nano, Vols 1-3*, pp. 988-991, 2007. [Online]. Available: <Go to ISI>://WOS:000252957300248.
- [3] H. Pedersen, S. Kozerke, S. Ringgaard, K. Nehrke, and W. Y. Kim, "k-t PCA: temporally constrained k-t BLAST reconstruction using principal component analysis," *Magn Reson Med*, vol. 62, no. 3, pp. 706-16, Sep 2009, doi: 10.1002/mrm.22052.
- [4] B. Zhao, J. P. Haldar, A. G. Christodoulou, and Z. P. Liang, "Image reconstruction from highly undersampled (k, t)-space data with joint partial separability and sparsity constraints," *IEEE Trans Med Imaging*, vol. 31, no. 9, pp. 1809-20, Sep 2012, doi: 10.1109/TMI.2012.2203921.
- [5] M. Fu *et al.*, "High-Resolution Dynamic Speech Imaging with Joint Low-Rank and Sparsity Constraints," *Magn. Reson. Med.*, vol. 73, no. 5, pp. 1820-1832, May 2015, doi: 10.1002/mrm.25302.

-
- [6] R. Otazo, E. Candes, and D. K. Sodickson, "Low-rank plus sparse matrix decomposition for accelerated dynamic MRI with separation of background and dynamic components," *Magn Reson Med*, vol. 73, no. 3, pp. 1125-36, Mar 2015, doi: 10.1002/mrm.25240.
- [7] B. Tremoulheac, N. Dikaios, D. Atkinson, and S. R. Arridge, "Dynamic MR Image Reconstruction–Separation From Undersampled (k, t)-Space via Low-Rank Plus Sparse Prior," *IEEE Trans Med Imaging*, vol. 33, no. 8, pp. 1689-1701, 2014.
- [8] J. He, Q. Liu, A. G. Christodoulou, C. Ma, F. Lam, and Z. P. Liang, "Accelerated High-Dimensional MR Imaging With Sparse Sampling Using Low-Rank Tensors," *IEEE Trans Med Imaging*, vol. 35, no. 9, pp. 2119-29, Sep 2016, doi: 10.1109/TMI.2016.2550204.
- [9] D. Wang, D. S. Smith, and X. Yang, "Dynamic MR image reconstruction based on total generalized variation and low-rank decomposition," *Magn Reson Med*, vol. 83, no. 6, pp. 2064-2076, Jun 2020, doi: 10.1002/mrm.28064.
- [10] K. H. Jin and J. C. Ye, "Sparse and Low-Rank Decomposition of a Hankel Structured Matrix for Impulse Noise Removal," (in English), *IEEE Trans. Image Process.*, vol. 27, no. 3, pp. 1448-1461, Mar 2018, doi: 10.1109/Tip.2017.2771471.
- [11] H. F. Li, X. G. He, D. P. Tao, Y. Y. Tang, and R. X. Wang, "Joint medical image fusion, denoising and enhancement via discriminative low-rank sparse dictionaries learning," (in English), *Pattern Recognit.*, vol. 79, pp. 130-146, Jul 2018, doi: 10.1016/j.patcog.2018.02.005.

Reviewer #1 (Remarks to the Author):

In the current revision of the manuscript, the authors have made several changes. The following comments are organized as follows: First, comments related to the points of critique that were previously addressed by the authors are listed and numbered according to the original comments. Subsequently, additional points of critique are presented, starting with number 6.

1. The authors claim that the proposed LSFP-Net is specially designed for intervention. However, in the LSFP modeling (Equation (1)), it is not explicitly indicated which terms are specifically designed for intervention.

5. In the comparative experiments, the authors chose different numbers of unfolding iterations for different methods. Why was a varying number of unfolding iterations selected, and what was the rationale behind these choices?

6. In Figure 2, L+S appears to outperform LSFP significantly. What could explain the substantial lag of L+S-Net behind LSFP-Net?

7. The paper does not discuss the parameter quantities of various methods in the comparative experiments. Conducting comparative experiments under varying parameter quantities may raise suspicions of unfairness.

Reviewer #2 (Remarks to the Author):

The revised version addressed all comments. The manuscript is well written and is recommended for publication.

Response to Reviewers

Reviewer #1:

In the current revision of the manuscript, the authors have made several changes. The following comments are organized as follows: First, comments related to the points of critique that were previously addressed by the authors are listed and numbered according to the original comments. Subsequently, additional points of critique are presented, starting with number 6.

Response: Thanks for your constructive critiques. We have carefully gone through all your comments and addressed them in our revised manuscript.

1. The authors claim that the proposed LSFP-Net is specially designed for intervention. However, in the LSFP modeling (Equation (1)), it is not explicitly indicated which terms are specifically designed for intervention.

Response: We apologize for the confusion. For clarity, we have modified the following sentences in the methods section before introducing Equation (1) (lines 348-352):

“For imaging during the intervention process, the background remains relatively stable, with changes primarily centered around the target region. Therefore, the low rankness of the background information is leveraged by using a low-rank matrix \mathbf{L} , and the sparsity of the intervention feature is incorporated through a sparse matrix \mathbf{S} . The image sequence \mathbf{x} could therefore be decomposed into a low-rank matrix \mathbf{L} and a sparse matrix \mathbf{S} , i.e., $\mathbf{x} = \mathbf{L} + \mathbf{S}$.”

The sentence in the discussion section was also modified (lines 277-279):

“The novelty of our method is that the LSFP-Net is specially designed for targeted intervention by leveraging the low rankness of the background information and the sparsity of the intervention feature. This makes it especially useful for real-time interventional guidance.”

5. In the comparative experiments, the authors chose different numbers of unfolding iterations for different methods. Why was a varying number of unfolding iterations selected, and what was the rationale behind these choices?

Response: Thank you for pointing this out. The CRNN was kept with its original settings as a reference. For deep learning (DL) algorithms utilizing low-rank constraints, initially, a total of 3 iterations were maintained for SLR-Net, L+S-Net, and LSFP-Net. However, to ensure a similar level of total parameters, adjustments were made since the parameters of LSFP-Net with 3 iterations were approximately 2 times that of ISTA-Net and SLR-Net, and 7 times that of L+S-Net. Consequently, we updated the results from ISTA-Net (12 iterations), SLR-Net (4 iterations), L+S-Net (15 iterations), and LSFP-Net (2 iterations) to achieve comparable total parameters. It is important to strike a balance between performance and computational time for the proposed LSFP-Net model. Comparative analysis with other DL-based methods revealed that LSFP-Net with 2 iterations achieved better performance while having a similar number of parameters (**Fig. 2**, **Fig. 3**, and **Supplementary Table 1**). Nevertheless, when the number of iterations was increased to 3, the PSNR/SSIM values improved with only a slight increase of 0.09s in computational time. Hence, for

the phantom and cadaveric experiments, LSFP-Net with 3 iterations was chosen as the optimal configuration. The following methods and results sections and figures were updated accordingly.

In the methods section (lines 431-440), we have added a paragraph explaining the comparison details mentioned above:

“Comparison with other algorithms

In this study, with 10 spokes for the reconstruction of each frame (acceleration factor $R=20$), we compared the results of LSFP-Net with two iterative CS methods (L+S²³ and LSFP³¹) and four DL-based methods (CRNN⁴⁰, ISTA-Net⁴⁴, SLR-Net⁴⁸, and L+S-Net⁴⁹). For detailed performance comparison using both the simulated and DBS datasets, CRNN was kept with its original settings as a reference. A similar number of model parameters were kept by varying unfolding iterations for the DL-based unrolled networks. Specifically, a total of 12, 4, 15, and 2 iterations were used for ISTA-Net, SLR-Net, L+S-Net, and LSFP-Net, respectively (**Supplementary Table 1**). Based on the comparative results (**Fig. 2**, **Fig. 3**, and **Supplementary Table 1**), LSFP-Net with 3 iterations was chosen for the phantom and cadaveric experiments by considering the tradeoff between reconstruction quality and computation time.”

In the results section (lines 131-144), the following paragraph was updated with comparisons with different iteration numbers:

“**Iterations and convolutional layers for LSFP-Net.** The numbers of iterative blocks N_b and convolutional layers N_c in $\{\psi_L, \psi_L^T, \psi_S, \psi_S^T\}$ determine the depth of LSFP-Net. To figure out their effect on the reconstruction, different values of N_b and N_c were investigated. First, different $N_b=1, 2, 3, 5, 7, 9$, and 11 were used with a fixed $N_c=3$ (**Fig. 3 (d)-(f)**). The PSNR/SSIM improved from 28.42/0.93 to 39.11/0.99 with the increase of iteration blocks from 1 to 11, and the increase of time cost from 0.11 s to 1.02 s. Therefore, a tradeoff is needed to balance performance and computational time. A comparison with other DL-based methods showed that LSFP-Net with $N_b=2$ achieved better performance with a similar number of parameters (**Fig. 2**, **Fig. 3**, and **Supplementary Table 1**). Moreover, when N_b was increased to 3, the PSNR/SSIM values improved with only a slight increase of 0.09s in computational time. As a result, $N_b=3$ was chosen for the phantom and cadaver experiments. Second, $N_c=3, 5, 7, 9$, and 11 were also used with a fixed $N_b=3$ (**Fig. 3 (g)-(i)**). The reconstruction quality fluctuated and the reconstruction time increased with increasing N_c . This was because LSFP-Net became more complex as the number of convolutional layers increased, which could affect the generalizability of the network. Therefore, $N_b=3$ and $N_c=3$ were selected for the reconstruction of phantom and cadaver experiments.”

A comparison of DL-based methods with varied numbers of iterations and parameters in simulated datasets of brain intervention is summarized in **Supplementary Table 1**.

Supplementary Table 1. A comparison of DL-based methods with different iterations in simulated datasets of brain intervention. The adopted iterations are in bold font.

Method	Iterations	Parameters ($\times 10^4$)	PSNR (dB)	SSIM	Time (s)
CRNN	/	29.7794	28.74\pm1.47	0.9255\pm0.0101	0.25
ISTA-Net	6	22.4652	25.96 \pm 1.24	0.8917 \pm 0.0128	0.08
	9	33.6978	27.28 \pm 1.56	0.8994 \pm 0.0143	0.09
	12	44.9304	28.04\pm0.96	0.9014\pm0.0077	0.12
	18	67.3956	29.63 \pm 1.39	0.9267 \pm 0.0085	0.17
SLR-Net	2	22.4648	28.88 \pm 1.61	0.9296 \pm 0.0083	0.10

	3	33.6973	28.14±1.60	0.9246±0.0114	0.16
	4	44.9298	29.22±1.73	0.9260±0.0109	0.20
	5	56.1625	29.24±1.51	0.9297±0.0117	0.23
	6	67.3948	29.28±2.20	0.9267±0.0123	0.29
L+S-Net	3	9.0725	29.65±1.96	0.9242±0.0110	0.18
	5	15.1209	30.95±2.08	0.9408±0.0102	0.24
	7	21.1693	31.84±2.37	0.9522±0.0088	0.34
	9	27.2177	32.71±2.15	0.9590±0.0071	0.41
	11	33.2661	33.55±2.19	0.9625±0.0072	0.49
	13	39.3145	34.30±2.05	0.9671±0.0049	0.57
	15	45.3629	34.88±2.05	0.9713±0.0044	0.67
LSFP-Net	1	22.4646	28.42±1.67	0.9264±0.0121	0.11
	2	44.9292	35.90±1.13	0.9705±0.0057	0.21
	3	67.3938	36.58±1.17	0.9755±0.0039	0.29
	5	112.3230	37.72±1.19	0.9830±0.0023	0.47
	7	157.2522	37.85±1.82	0.9838±0.0031	0.66
	9	202.1814	37.58±1.79	0.9849±0.0024	0.83
	11	247.1106	39.11±1.41	0.9879±0.0018	1.02

6. In Figure 2, L+S appears to outperform LSFP significantly. What could explain the substantial lag of L+S-Net behind LSFP-Net?

Response: Thank you for pointing this out. We carefully checked the regularized parameters of LSFP and L+S and updated the results (**Figs. 2 and 4**). Compared to L+S, LSFP exploited more priors (i.e., the spatial sparsity of both low-rank and sparse components) but introduced more regularization parameters (λ_L and λ_S in L+S model VS. λ_L , λ_S , λ_L^ψ , and λ_S^ψ in LSFP model). By unrolling LSFP into a deep neural network, fewer iterations are needed and the regularized parameters are learnable during the LSFP-Net training, avoiding ad hoc parameter tuning yet achieving near-optimal performance. For clarity, the following sentences have been added to the results section (lines 121-124):

“The enhanced performance of LSFP-Net can be attributed to its ability to leverage the spatial sparsity of both low-rank and sparse components, as well as its capability to learn the regularized parameters during the training stage, setting it apart from other methods.”

7. The paper does not discuss the parameter quantities of various methods in the comparative experiments. Conducting comparative experiments under varying parameter quantities may raise suspicions of unfairness.

Response: Thank you. The CRNN was kept with its original settings as a reference. For deep learning (DL) algorithms utilizing low-rank constraints, initially, a total of 3 iterations were maintained for SLR-Net, L+S-Net, and LSFP-Net. However, to ensure a similar level of total parameters,

adjustments were made since the parameters of LSFP-Net with 3 iterations were approximately 2 times that of ISTA-Net and SLR-Net, and 7 times that of L+S-Net. Consequently, we updated the results from ISTA-Net (12 iterations), SLR-Net (4 iterations), L+S-Net (15 iterations), and LSFP-Net (2 iterations) to achieve comparable total parameters. It is important to strike a balance between performance and computational time for the proposed LSFP-Net model. Comparative analysis with other DL-based methods revealed that LSFP-Net with 2 iterations achieved better performance while having a similar number of parameters (**Fig. 2, Fig. 3, and Supplementary Table 1**). Nevertheless, when the number of iterations was increased to 3, the PSNR/SSIM values improved with only a slight increase of 0.09s in computational time. Hence, for the phantom and cadaveric experiments, LSFP-Net with 3 iterations was chosen as the optimal configuration. The following methods and results sections and figures were updated accordingly.

In the methods section (lines 431-440), we have added a paragraph explaining the comparison details mentioned above:

“Comparison with other algorithms

In this study, with 10 spokes for the reconstruction of each frame (acceleration factor $R=20$), we compared the results of LSFP-Net with two iterative CS methods (L+S²³ and LSFP³¹) and four DL-based methods (CRNN⁴⁰, ISTA-Net⁴⁴, SLR-Net⁴⁸, and L+S-Net⁴⁹). For detailed performance comparison using both the simulated and DBS datasets, CRNN was kept with its original settings as a reference. A similar number of model parameters were kept by varying unfolding iterations for the DL-based unrolled networks. Specifically, a total of 12, 4, 15, and 2 iterations were used for ISTA-Net, SLR-Net, L+S-Net, and LSFP-Net, respectively (**Supplementary Table 1**). Based on the comparative results (**Fig. 2, Fig. 3, and Supplementary Table 1**), LSFP-Net with 3 iterations was chosen for the phantom and cadaveric experiments by considering the tradeoff between reconstruction quality and computation time.”

In the results section (lines 131-144), the following paragraph was updated with comparisons with different iteration numbers:

“**Iterations and convolutional layers for LSFP-Net.** The numbers of iterative blocks N_b and convolutional layers N_c in $\{\psi_L, \psi_L^T, \psi_S, \psi_S^T\}$ determine the depth of LSFP-Net. To figure out their effect on the reconstruction, different values of N_b and N_c were investigated. First, different $N_b=1, 2, 3, 5, 7, 9,$ and 11 were used with a fixed $N_c=3$ (**Fig. 3 (d)-(f)**). The PSNR/SSIM improved from 28.42/0.93 to 39.11/0.99 with the increase of iteration blocks from 1 to 11, and the increase of time cost from 0.11 s to 1.02 s. Therefore, a tradeoff is needed to balance performance and computational time. A comparison with other DL-based methods showed that LSFP-Net with $N_b=2$ achieved better performance with a similar number of parameters (**Fig. 2, Fig. 3, and Supplementary Table 1**). Moreover, when N_b was increased to 3, the PSNR/SSIM values improved with only a slight increase of 0.09s in computational time. As a result, $N_b=3$ was chosen for the phantom and cadaver experiments. Second, $N_c=3, 5, 7, 9,$ and 11 were also used with a fixed $N_b=3$ (**Fig. 3 (g)-(i)**). The reconstruction quality fluctuated and the reconstruction time increased with increasing N_c . This was because LSFP-Net became more complex as the number of convolutional layers increased, which could affect the generalizability of the network. Therefore, $N_b=3$ and $N_c=3$ were selected for the reconstruction of phantom and cadaver experiments.”

Reviewer #2:

The revised version addressed all comments. The manuscript is well written and is recommended for publication.

Response: We sincerely thank the reviewer for your time and for recognizing our work.

Reviewer #1 (Remarks to the Author):

The authors have addressed all my concerns.

Response to Reviewers

Reviewer #1:

The authors have addressed all my concerns.

Response: We sincerely thank the reviewer for your time and for recognizing our work.